# *Delilah*, *prospero*, and *D-Pax2* constitute a gene regulatory network essential for the development of functional proprioceptors

**Adel Avetisyan, Yael Glatt, Maya Cohen, Yael Timerman, Nitay Aspis, Atalya Nachman, Naomi Halachmi, Ella Preger-Ben Noon\*, Adi Salzberg\***

Department of Genetics and Developmental Biology, The Rappaport Faculty of Medicine and Research Institute, Technion-Israel Institute of Technology, Haifa, Israel

**Abstract** Coordinated animal locomotion depends on the development of functional proprioceptors. While early cell-fate determination processes are well characterized, little is known about the terminal differentiation of cells within the proprioceptive lineage and the genetic networks that control them. In this work we describe a gene regulatory network consisting of three transcription factors–Prospero (Pros), D-Pax2, and Delilah (Dei)–that dictates two alternative differentiation programs within the proprioceptive lineage in *Drosophila*. We show that D-Pax2 and Pros control the differentiation of cap versus scolopale cells in the chordotonal organ lineage by, respectively, activating and repressing the transcription of *dei*. Normally, D-Pax2 activates the expression of *dei* in the cap cell but is unable to do so in the scolopale cell where Pros is co-expressed. We further show that D-Pax2 and Pros exert their effects on *dei* transcription via a 262 bp chordotonal-specific enhancer in which two D-Pax2- and three Pros-binding sites were identified experimentally. When this enhancer was removed from the fly genome, the cap- and ligament-specific expression of *dei* was lost, resulting in loss of chordotonal organ functionality and defective larval locomotion. Thus, coordinated larval locomotion depends on the activity of a *dei* enhancer that integrates both activating and repressive inputs for the generation of a functional proprioceptive organ.

**\*For correspondence:**
pregere@technion.ac.il (EP-BN);
adis@technion.ac.il (AS)

**Competing interest:** The authors declare that no competing interests exist.

## Editor's evaluation

The study provides compelling evidence for a gene regulatory network involved in generating different sensory cell types from a common progenitor. The careful work shows how an enhancer can integrate the antagonistic relationship between two transcription factors for *Drosophila* sensory system development.

## Introduction

A central question in developmental biology is how different cells that originate in the same lineage and develop within the same organ, acquire unique identities, properties and specialized morphologies. One of the common mechanisms involved in cell fate diversification within a cell lineage is asymmetric cell division in which cytoplasmic determinants of the mother cell differentially segregate into one of the two daughter cells. This asymmetry is then translated into differential gene expression and the activation of cell-type-specific gene regulatory networks (GRN) that dictate the differentiation programs of cells with unique properties. The transition from a primary cell fate to the characteristic phenotype of a fully differentiated cell involves complex GRNs in which numerous genes regulate

each other's expression. Despite this complexity, genetic analyses in well-characterized developmental systems can often reveal elementary interactions in small GRNs which dictate a specific cell fate, or a specific feature of the differentiating cell.

Many of the core components and the central processes underlying asymmetric cell divisions and primary cell fate decisions have been uncovered in studies performed on the central and peripheral nervous system (PNS) of *Drosophila* (e.g. *Knoblich, 2008*; *Schweisguth, 2015*). The PNS of *Drosophila* contains two classes of multicellular sensory organs, external sensory organs and chordotonal organs (ChOs), whose lineages share a similar pattern of asymmetric cell divisions (*Lai and Orgogozo, 2004*). In both types of organs, the neuron and support cells, which collectively comprise the sensory organ, arise from a single sensory organ precursor cell (SOP) through a sequence of precisely choreographed asymmetric cell divisions. Antagonistic interactions involving Notch and Numb are key regulators of the asymmetry generated between each two sibling cells within these lineages (*Gönczy, 2008*; *Rebeiz et al., 2011*; *Reeves and Posakony, 2005*). Unlike the primary cell-fate specification, which has been extensively investigated, the process of terminal differentiation of the post-mitotic progeny is poorly understood. We are using the larval lateral pentascolopidial ChO (LCh5) as a model system to study cell fate diversification within a sensory lineage.

The LCh5 organ is composed of five mechano-sensory units (scolopidia) that are attached to the cuticle via specialized epidermal attachment cells. Each of the five scolopidia originates in a single precursor cell that divides asymmetrically to generate five of the six cell types that construct the mature organ: the neuron, scolopale, ligament, cap, and cap-attachment cell (*Brewster and Bodmer, 1995*; *Figure 1A–C*). Three of the five cap-attachment cells are rapidly removed by apoptosis, leaving two cap-attachment cells that anchor the five cap cells to the epidermis (*Avetisyan and Salzberg, 2019*). Later in development, following the migration of the LCh5 organ from the dorsal to the lateral region of the segment, a single ligament-attachment cell is recruited from the epidermis to anchor the five ligament cells to the cuticle (*Inbal et al., 2004*). The mature LCh5 organ responds to mechanical stimuli generated by muscle contractions that lead to relative displacement of the attachment cells and the consequent shortening of the organ (*Hassan et al., 2019*).

Very little is known about the unique cell-type-specific differentiation programs that characterize each of the ChO cells, whose morphologies and mechanical properties differ dramatically from each other. To address this issue, we focus on the transcription factor Taxi wings/Delilah (Dei), an important regulator of cell adhesion (*Egoz-Matia et al., 2011*), which is expressed in the four accessory cell types (cap, ligament, cap-attachment and ligament-attachment) but is excluded from the neuron and the scolopale cell. Even though *dei* is expressed in all four accessory cells, its expression in these cells is differentially regulated. The transcription of *dei* in the ChO is controlled by two *cis*-regulatory modules (CRMs): The *dei*$^{attachment}$ enhancer, located ~2.5 Kb upstream of the *dei* transcription start site, drives expression in the cap-attachment and ligament-attachment cells (as well as tendon cells), whereas the *dei*$^{ChO-1353}$ enhancer, an intronic 1353 bp DNA fragment, drives *dei* expression specifically in the cap and ligament cells (*Nachman et al., 2015*; *Figure 1D–F*). The *dei*$^{attachment}$ enhancer was shown to be activated by the transcription factor Stripe, which is considered a key regulator of tendon cell development (*Becker et al., 1997*) and a known determinant of attachment cell identity (*Klein et al., 2010*). Here, we provide a high-resolution dissection of the *dei*$^{ChO-1353}$ enhancer and show that it integrates both activating and repressive cues to drive *dei* expression in cap and ligament cells while suppressing it in scolopale cells. We find that D-Pax2/Shaven (Sv), which is expressed in both branches of the cell lineage, is a positive regulator of *dei* and that Prospero (Pros) inhibits *dei* expression specifically in the scolopale cell. This small GRN is required for the realization of differentiation programs characterizing cap versus scolopale cell fates and is therefore essential for ChO functionality and coordinated larval locomotion.

## Results

### Identifying D-Pax2/Sv as a putative direct activator of *dei* expression

To reveal the gene network that regulates *dei* expression in the ChO cells, we used the previously identified ChO-specific *dei*$^{ChO-1353}$ enhancer or the shorter version of it - *dei*$^{ChO-389}$, both described in *Nachman et al., 2015*; *Figure 1D and F–G*, as an entry point. As part of this work, we have further narrowed down the critical regulatory region to an evolutionarily conserved 262 bp fragment,

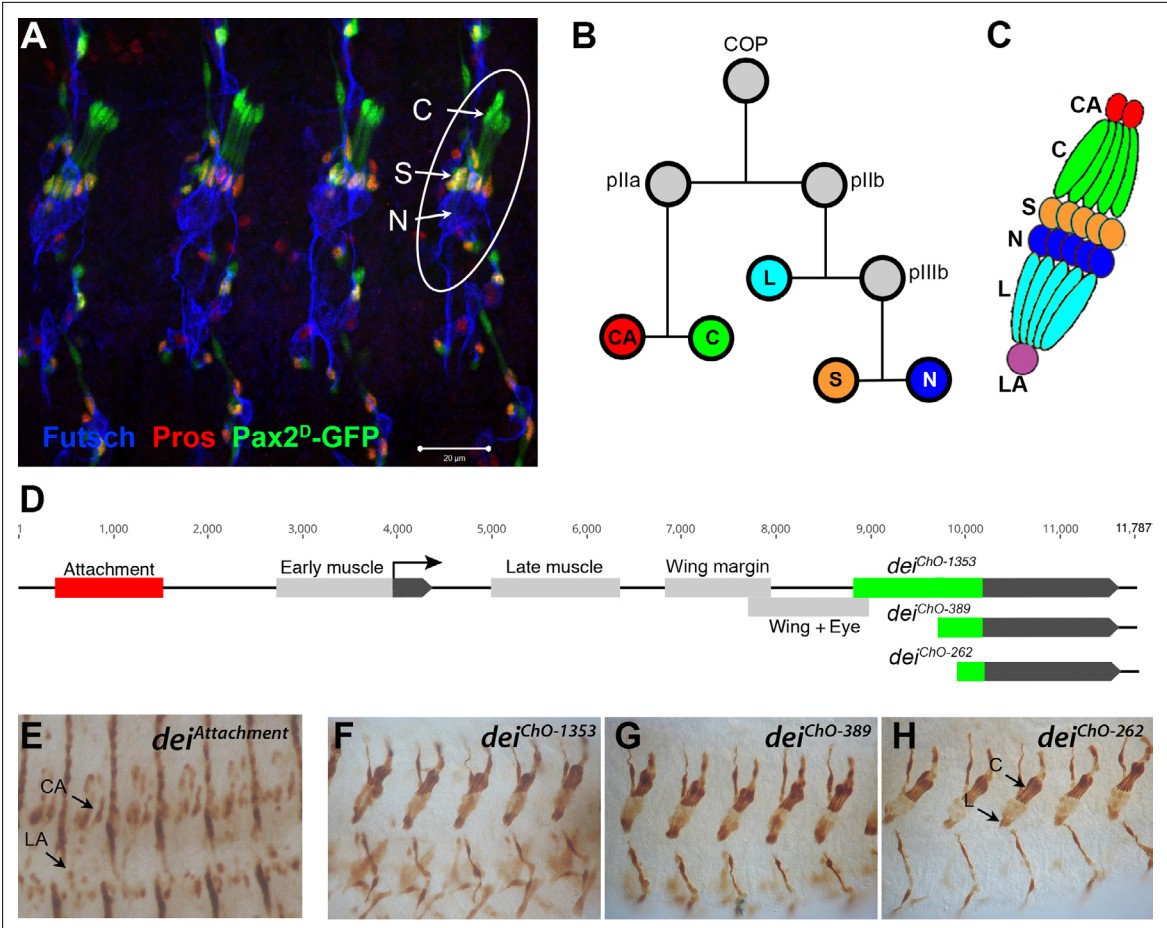

**Figure 1.** The LCh5 organ and the *dei* gene. (**A**) Four abdominal segments of a stage 16 embryo carrying the *sv/Pax2^{D1}-GFP* reporter (green) which labels the cap and scolopale cells, stained with anti-Pros (red) which labels the nuclei of scolopale cells, and anti-Futsch (blue), which labels the neurons. One LCh5 organ is circled and the cap cells (**C**), scolopale cells (**S**) and neurons (**N**) are indicated. The cap-attachment, ligament-attachment and ligament cells are not stained. Scale bar = 20 µm. (**B–C**) the ChO lineage (**B**) and schematic illustration of an LCh5 organ (**C**). The neurons are depicted in blue, the scolopale cells in orange, the ligament cells in cerulean, the cap cells in green, the cap-attachment cells in red and the ligament-attachment cell in purple. (**D**) Schematic representation of the *dei* locus showing the two exons (black boxes) and the *cis* regulatory modules (CRM) that drive gene expression within the ChO lineage: the *dei^{attachment}* CRM which drives expression in the attachment cells (red box) and the *dei^{ChO}* CRM which drives expression in the cap and ligament cells (green box). The *dei^{ChO}* CRM was originally mapped to a 1353 bp fragment (*dei^{ChO-1353}*) and was then delimited to a smaller 389 bp fragment located immediately upstream to the second exon (*dei^{ChO-389}*). As part of this work, the ChO-specific CRM was further delimited to a 262 bp fragment (*dei^{ChO-262}*). (**E–H**) The embryonic expression patterns driven by the *dei^{attachment}* (**E**), *dei^{ChO-1353}* (**F**), *dei^{ChO-389}* (**G**), and *dei^{ChO-262}* (**H**) enhancers are shown. C, cap cell; L, ligament cell; CA, cap-attachment cell, LA, ligament-attachment cell.

The online version of this article includes the following figure supplement(s) for figure 1:

**Figure supplement 1.** Identifying potential direct regulators of *dei* in the ChO using a Y1H screen.

*dei^{ChO-262}*, which drives ChO-specific expression in a pattern indistinguishable from the expression pattern driven by the larger fragments *dei^{ChO-1353}* and *dei^{ChO-389}* (*Figure 1D, F and H*). These regulatory fragments were used in two types of screens aimed at identifying genes that regulate the expression of *dei* in the ChO lineage. The first screen was an RNAi-based phenotypic screen conducted in larvae (described in *Hassan et al., 2018*). It capitalized on a reporter fly strain in which the cap and ligament cells expressed green fluorescent protein (GFP) under the regulation of the *dei^{ChO-1353}* enhancer, while the attachment cells expressed red fluorescent protein (RFP) under the regulation of the *dei^{attachment}* enhancer (*Halachmi et al., 2016*). One of the 31 genes identified in that screen as being required for normal morphogenesis of the larval LCh5 organ was *shaven* (*sv*), which encodes for the *Drosophila* Pax2 homologue D-Pax2. The knockdown of *sv* within the ChO lineage led to loss of expression of the *dei^{ChO-1353}* reporter from the cap cells, suggesting that D-Pax2 is a positive regulator of *dei* transcription in this cell type (*Hassan et al., 2018*).

Here, we describe the second screen, a yeast one hybrid (Y1H) screen, aimed at identifying proteins that bind *directly* to the *dei*$^{ChO-389}$ enhancer. The screen was performed by Hybergenics Services on a *Drosophila* whole embryo library (ULTImate Y1H screen *vs Drosophila* Whole Embryo RP2 0–12 + 12-24 hr), using the *dei*$^{ChO-389}$ sequence as a bait. Screening was performed using the Aureobasidin A selection system (described in detail in the Materials and methods section). Out of 124 million clones screened, 146 clones that were found to grow on the selective medium containing 400 ng/ml of the yeast antibiotic agent Aureobasidin-A were sequenced (*Supplementary file 1*). Two proteins were identified to bind the bait with very high confidence in the interaction: D-Pax2/Sv (eight independent clones) and LamC (21 independent clones). Three additional candidates were identified as binders with moderate confidence in the interaction: Fax (one clone), Lola (two independent clones), and Toy (three independent clones) (*Figure 1—figure supplement 1* and *Supplementary file 1*). Together, the results of the two independent screens identified D-Pax2/Sv as a putative direct transcriptional activator of *dei* and an important player in ChO development.

## D-Pax2/Sv activates *dei* expression in the cap cell

To validate the *sv* RNAi-induced phenotype, we characterized the LCh5 organs of *sv* mutant embryos (a null allele - *sv*$^6$). In accordance with the knockdown phenotypes, the expression of *dei* in the cap cell was abolished in *sv*-deficient embryos (*Figure 2A–B*). The expression of additional genes associated with proper differentiation of the cap cell, such as *αTub85E*, was reduced as well (*Figure 2C–D*). The loss of *sv* did not eliminate the expression of either scolopale-specific proteins (Crumbs, Pros) or neuronal markers (Futsch, Elav, Nrg), suggesting that it did not affect primary cell fate decisions, however, its loss prevented normal morphogenesis of the sensory unit (*Figure 2E–H*).

To further test the ability of D-Pax2/Sv to activate *dei* expression in vivo, we ectopically expressed *sv* under the regulation of *en-Gal4* and examined the resulting changes in gene expression patterns. The ectopic expression of *sv* led to ectopic expression of both the endogenous *dei* gene and the *dei*$^{ChO-262}$ transcriptional reporter, as well as the cap cell marker αTub85E (*Figure 2I–L*). The ectopic expression of D-Pax2 in the *en* domain had detrimental effects on the pattern of ChO migration and possibly on other morphogenetic processes in the embryo. Thus, the identification of cells based on their position was not feasible in these embryos. However, this experiment clearly showed that ectopic expression of D-Pax2 could activate expression of the *dei* gene and the *dei*$^{ChO-262}$-*lacZ* reporter in epidermal cells within the *en* domain. Thus, D-Pax2/Sv ability to activate *dei* is not restricted to the ChO lineage, or the peripheral nervous system in general. Altogether, these observations corroborate the notion that D-Pax2/Sv is an activator of *dei* that plays a critical role in ChO morphogenesis.

## Pros represses *dei* in the scolopale cell

The *sv* gene is expressed broadly within the ChO lineage during early stages of organ development and is then gradually restricted to the cap and scolopale cells (*Czerny et al., 1997*; *Fu et al., 1998*; *Fu and Noll, 1997*), where its expression level remains high during late embryogenesis and larval stages. Yet, the expression of its downstream target gene *dei* is normally activated in the cap cell but is excluded from the scolopale cell. This discrepancy in the expression pattern of D-Pax2/Sv and *dei* may point to the presence of a scolopale-specific repressor that prevents the transcription of *dei* in this D-Pax2/Sv-expressing cell. The results of the abovementioned RNAi screen identified the transcription factor Pros as a good candidate for being that repressor. *pros* and *sv* RNAi had opposite effects on the expression of the *dei*$^{ChO-1353}$-*GFP* reporter. Although the knockdown of *sv* led to a loss of the reporter expression from the cap cell, the knockdown of *pros* led to expansion of its expression into the scolopale cell suggesting that, normally, Pros represses *dei* in this cell (*Hassan et al., 2018*).

To validate the *pros* knockdown phenotypes and to further test the idea that Pros acts as a repressor of *dei* in the scolopale cell, we characterized the ChOs of *pros*$^{17}$ embryos. The loss of *pros* led to ectopic expression of both the endogenous *dei* gene and the *dei*$^{ChO-262}$ reporter, as well as the cap cell marker αTub85E in the scolopale cells of the LCh5 organs (*Figure 3A–B*). Similar effect was evident in the scolopale cells of the LCh1 and VChA/B organs (*Figure 3—figure supplement 1*). The *pros*-deficient scolopale cells developed into cap-like cells rather than ligament-like or attachment cell-like cells. This was indicated by the upregulation of *dei* which was not accompanied by upregulation of the transcription factor Sr that is normally co-expressed with Dei in the ligament and attachment cells but is excluded from the cap cells (*Figure 3C–D*). The observed alterations in gene expression pattern

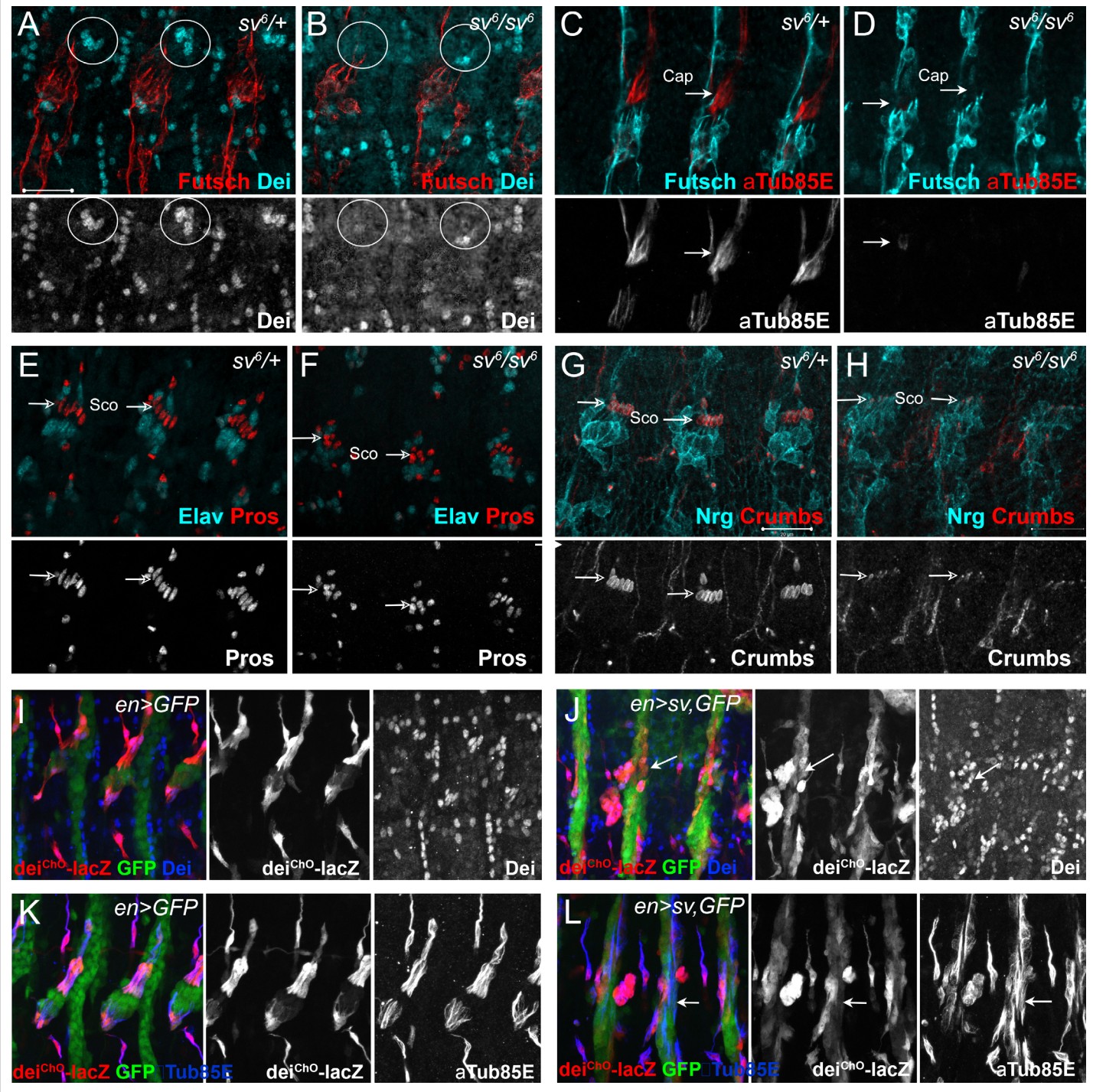

**Figure 2.** Sv/D-Pax2 activates *dei* expression in the cap cell. (A–B) Representative abdominal segments of stage 16 embryos stained for Dei (cyan) and the neuronal marker Futsch (red). The anti-Dei staining is shown separately in the lower panel. (A) A $sv^6$ heterozygous embryo demonstrating the expression of Dei in the cap cell nuclei (circled). (B) The expression of Dei is lost in homozygous $sv^6$ embryos. (C–D) Heterozygous (C) and homozygous (D) $sv^6$ embryos stained for αTub85E (red) and Futsch (cyan). The arrows mark the cap cells. The anti-αTub85E staining is shown separately in the lower panel. (E–F) Heterozygous (E) and homozygous (F) $sv^6$ embryos stained for Elav (cyan) and Pros (red). The arrows mark the nuclei of scolopale cells. The anti-Pros staining is shown separately in the lower panel. (G–H) Heterozygous (G) and homozygous (G) $sv^6$ embryos stained for Nrg (cyan) and Crumbs (red). The arrows mark the nuclei of scolopale (Sco) cells. The anti-Crumbs staining is shown separately in the lower panel. (I–L) Representative abdominal segments of stage 16 embryos that express GFP (I, K) or GFP and *sv* (J, L) under the regulation of *en-Gal4*. The embryos carry the $dei^{ChO-262}$-*lacZ* marker (anti bGal staining is shown in red) and are stained with anti-Dei (blue in I-J) or anti-αTub85E (blue in K-L). The staining patterns of $dei^{ChO-262}$-lacZ, Dei and αTub85E are shown separately on the right. Note the ectopic expression of Dei, $dei^{ChO-262}$-lacZ and αTub85E in epidermal cells within the *en* domain (arrows in J, L).

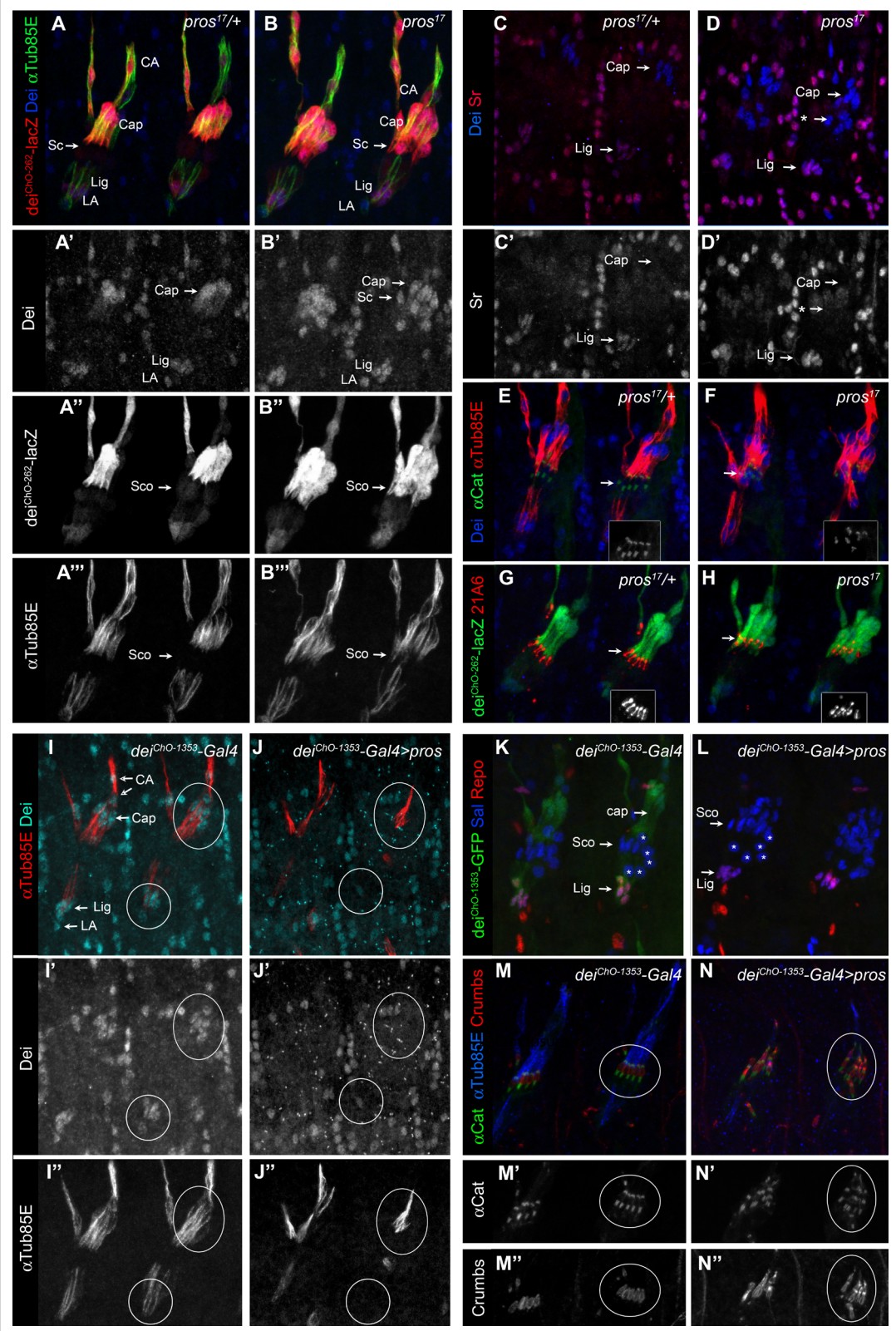

**Figure 3.** Pros represses *dei* in the scolopale cell. (**A–B**) Representative abdominal segments of stage 16 *pros¹⁷* heterozygous (**A**) and homozygous (**B**) embryos carrying the *dei^ChO-262^-lacZ* marker (shown in red) and stained for Dei (blue) and αTub85E (green). Each of the three channels is shown separately below. Note the expansion of Dei and the *dei^ChO-262^-lacZ* marker into the scolopale cells in *pros¹⁷* mutant embryos. The arrows point to the scolopale cells. (**C–D**) *pros¹⁷* heterozygous (**C**) and homozygous (**D**) embryos stained for Dei (blue) and Sr (red). Note that the ectopic expression of

*Figure 3 continued on next page*

*Figure 3 continued*

Dei in the scolopale cells of pros mutant embryo is not accompanied by ectopic expression of Sr (the arrow labeled with asterisks in D). (**E–F**) *pros*[17] heterozygous (**E**) and homozygous (**F**) embryos carrying an α-Catenin-GFP reporter and stained for Dei (blue, shown separately in the inset) and αTub85E (red). (**G–H**) *pros*[17] heterozygous (**G**) and homozygous (**H**) embryos carrying the *dei*[ChO-262]-lacZ reporter (shown in green) and stained with the scolopale marker anti-Eyes Shut (MAb21A6, red, shown separately in the inset). Note that the expression of both Eyes Shut/21A6 and α—Catenin is maintained in the *pros*-deficient scolopale cells. (**I–J**) *dei*[ChO-1353]-Gal4 (**I**) and *dei*[ChO-1353]> UAS pros (**J**) embryos stained for Dei (cyan, shown separately in I'-J') and αTub85E (red, shown separately in I"-J"). The cap and ligament cells are circled. Note the loss of Dei and αTub85E expression upon Pros expression. (**K–L**) *dei*[ChO-1353] (**K**) and *dei*[ChO-1353]> UAS pros (**L**) embryos carrying the *dei*[ChO-1353]-lacZ reporter (shown in green) stained for Sal (blue) and Repo (red). The cap, scolopale (Sco) and ligament (Lig) cells are indicated; the asterisks mark oenocyte cell nuclei. (**M–N**) *dei*[ChO-1353]-Gal4 (**M**) and *dei*[ChO-1353]> UAS pros (**N**) embryos carrying an α—Catenin-GFP reporter (green, shown separately in M'-N') and stained for Crumbs (red, shown separately in M"-N") and αTub85E (blue). Note the duplication of scolopale-specific structures in the cap cell expressing Pros ectopically (circled).

The online version of this article includes the following figure supplement(s) for figure 3:

**Figure supplement 1.** The loss of *pros* affects similarly the various types of larval ChOs.

**Figure supplement 2.** The number of neurons and ligament cells remain normal in *pros* mutant embryos.

do not reflect a full scolopale-to-cap cell fate transformation, as the Pros-deficient scolopale cells still maintain some of their scolopale-specific characteristics, such as Eyes Shut and α—Catenin expression (*Figure 3E–H*). As the number of neurons, ligament, cap, and cap-attachment cells remained normal in *pros* mutant embryos (*Figure 3—figure supplement 2*), we conclude that Pros does not influence primary cell-fate decisions within the LCh5 lineage.

The ability of Pros to repress *dei* was not restricted to the scolopale cell. Ectopic expression of *pros* in the LCh5 lineage, using a *dei*[ChO-1353]-Gal4 driver, abolished the expression of both the endogenous *dei* gene and a *dei*[ChO-1353]-GFP reporter in the cap and ligament cells as well (*Figure 3I–L*). In parallel to the repression of *dei*, ectopic expression of *pros* was sufficient for upregulating scolopale-specific genes and, moreover, for driving the formation of ectopic scolopale-specific structures within the affected cap cells. Most notably, the Pros-expressing cap cells manifested scolopale rods and ectopically expressed Crumbs and α-Catenin in a scolopale-characteristic pattern (*Figure 3M–N*).

## D-Pax2/Sv and Pros regulate the transcription of *dei* via *dei*[ChO-262]

The fact that the expression of the *dei*[ChO] transcriptional reporters was affected similarly to the endogenous *dei* gene by both *sv* and *pros* loss- and gain-of-function, suggested that both D-Pax2/Sv and Pros regulate *dei*'s transcription in the ChOs through this regulatory module. To test this hypothesis, we deleted the *dei*[ChO-262] region from the fly genome by CRISPR/Cas9-mediated genome editing, resulting in a new regulatory allele of *dei* (*dei*[ΔChO]) (*Figure 4—figure supplement 1*). To verify that the deletion of this intronic enhancer does not affect splicing of the transcript, cDNA was synthesized from homozygous *dei*[ΔChO] and control flies. A 416 bp fragment was amplified by PCR from the cDNA samples using primers located on both sides of the intron (in the 1st and 2nd exons) (*Figure 4—figure supplement 1B* and Materials and methods). Sequencing of the PCR products verified the presence of normally structured *dei* transcript in *dei*[ΔChO] mutants (*Figure 4—figure supplement 1C*).

As expected, in homozygous *dei*[ΔChO] embryos the expression of *dei* was lost from the cap and ligament cells but remained intact in the rest of the *dei*-expressing cells: CA, LA and tendon cells (*Figure 4A–B*). This observation strongly suggests that the *dei*[ChO-262] enhancer constitutes the sole regulatory module driving *dei* expression in the cap and ligament cells.

To further test the notion that D-Pax2/Sv regulates *dei* expression via the *dei*[ChO-262] enhancer we examined the ability of D-Pax2/Sv to activate *dei* expression in the *dei*[ΔChO] background. We found that ectopic expression of D-Pax2/Sv failed to induce ectopic *dei* expression in the *dei*[ΔChO] background (Compare *Figures 4C and 2L*), indicating that the *dei*[ChO-262] enhancer is indispensable for the ability of D-Pax2/Sv to activate *dei* transcription. In a complementary experiment, we tested the effect of *pros* loss-of-function on the expression of *dei* in the *dei*[ΔChO] background. We found that the *dei*[ΔChO] regulatory mutation was epistatic to *pros* loss-of-function, so that no ectopic expression of *dei* was observed in the *pros*-deficient scolopale cells in embryos homozygous for the *dei*[ΔChO] mutation (Compare *Figure 4D–E* and *Figure 3B and D*). Based on these results, we concluded that the expression of *dei* in ChOs depends on the presence of the *dei*[ChO-262] regulatory region and that this enhancer integrates the positive and negative inputs of D-Pax2/Sv and Pros, respectively, to drive a lineage-specific *dei* expression.

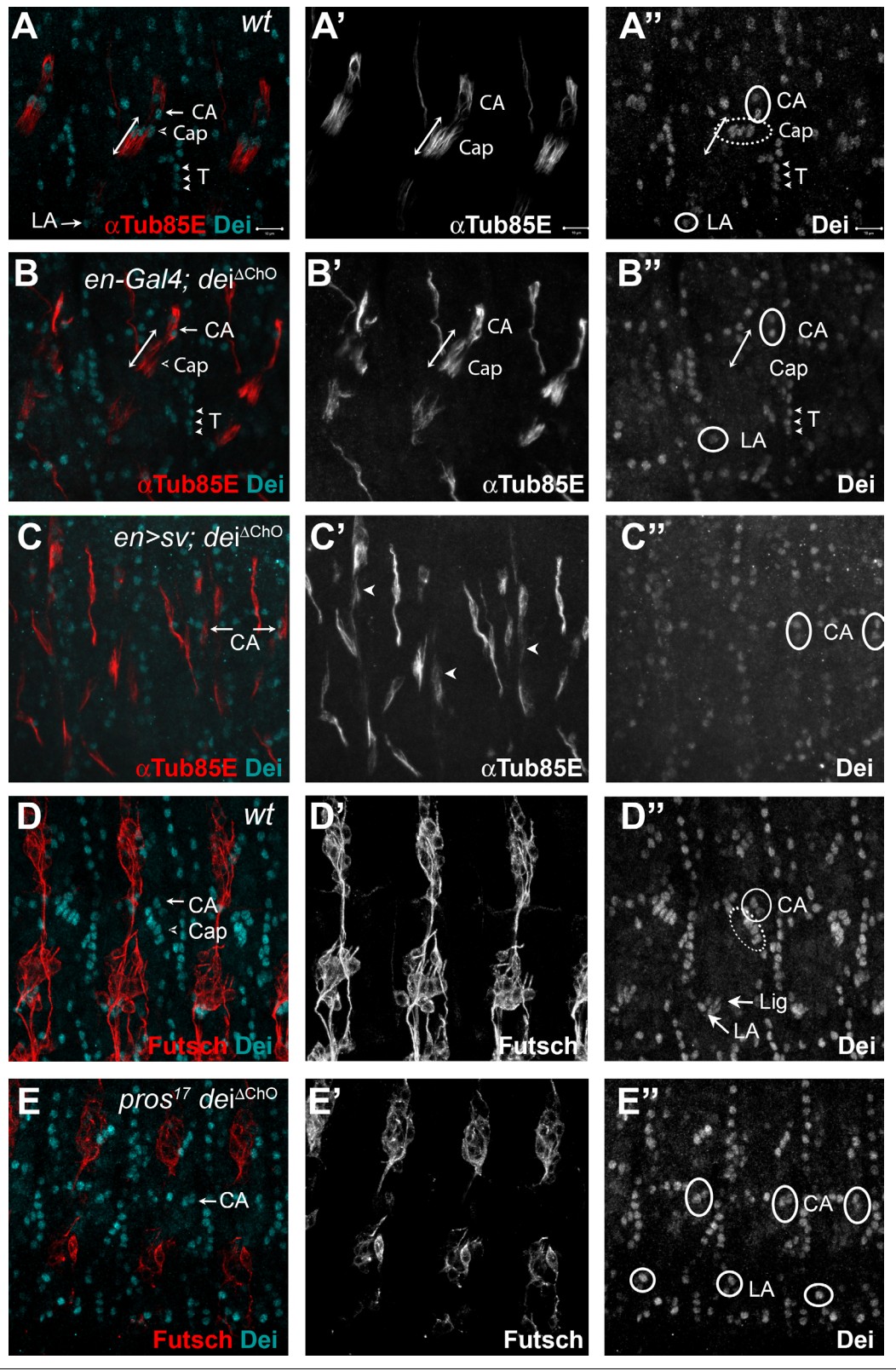

**Figure 4.** D-Pax2/Sv and Pros regulate the transcription of *dei* via the *dei^{ChO-262}* regulatory module. (**A–B**) Stage 16 *wt* (**A**) and *dei^{ΔChO}* (**B**) embryos stained for Dei (cyan) and αTub85E (red). Note that in (**B**) Dei is still evident in the cap-attachment and ligament-attachment cells (CA and LA, arrows in A, circled in A") , as well as in tendon cells (T; arrowheads), but is lost from the cap and ligament cells. The double edge arrows in A and B demarcate the length

*Figure 4 continued on next page*

*Figure 4 continued*

of the cap cells. (**C**) An embryo in which *sv* was expressed under the regulation of *en-Gal4* in a *dei^ΔChO* background. Sv was unable to induce Dei expression in the cap and ligament cells in the absence of the *dei^ChO-262* enhancer. Ectopic expression of αTub85E is evident within the *en* domain (arrowheads in C'). (**D–E**) *wt* (**D**) and A *pros^17 dei^ΔChO* homozygous embryo (**E**) stained for Dei (cyan) and anti-Futsch (red). The *pros^17 dei^ΔChO* embryo (**E**) presents a *dei^ΔChO*-like and not *pros^17*-like expression pattern of the Dei protein, indicating that the *dei^ΔChO* deletion is epistatic to *pros* loss-of-function. The neurons present the typical *pros* axonal pathfinding defects.

The online version of this article includes the following figure supplement(s) for figure 4:

**Figure supplement 1.** Deletion of the *dei^ChO262* enhancer does not affect splicing.

## D-Pax2/Sv and Pros are direct transcriptional regulators of *dei*

Our genetic analyses revealed that D-Pax2/Sv and Pros regulate the expression of *dei* in opposing manners through the function of the *dei^ChO-262* enhancer. In addition, the Y1H screen identified a direct interaction between D-Pax2/Sv and *dei^ChO-389* (which includes the *dei^ChO-262* module). We therefore hypothesized that the *dei^ChO-262* enhancer contains binding sites for D-Pax2/Sv and Pros. Motif search analysis predicted that the *dei^ChO-262* sequence encodes one canonical binding site for D-Pax2/Sv (*Figure 5B*, D-Pax2/Sv Site 2). On the other hand, we were unable to predict binding sites for Pros in the *dei^ChO-262* sequence.

To test whether D-Pax2/Sv actually binds to the predicted site, and to search for additional D-Pax2/Sv binding sites and for Pros binding sites, we systematically screened the *dei^ChO-262* sequence with electrophoretic mobility shift assays (EMSAs, *Figure 5* and *Figure 5—figure supplements 1 and 2*). As shown in *Figure 5C*, purified D-Pax2/Sv DNA-binding domain bound strongly to the region containing the predicted binding site (probe 7). In addition, we identified another region (in probe 3), that bound D-Pax-2/Sv at a lower affinity (*Figure 5C* and *Figure 5—figure supplement 1*). Using purified Pros-S DNA-binding domain we found that four fragments–probes 3, 4, 5, and 6–bound Pros in vitro (*Figure 5D*). Comprehensive mutagenesis and competition assays revealed three Pros conserved binding sites within these fragments (*Figure 5—figure supplement 2*), one of them partially overlapping the low-affinity binding site of D-Pax2/Sv (*Figure 5B* and *Figure 5—figure supplement 3*). Interestingly, none of these identified binding sites resemble the known binding sites for Pros identified by target detection assay (TDA) (*Hassan et al., 1997*), by functional studies (*Cook et al., 2003*), or by single-cell omics analyses (*Bravo González-Blas et al., 2020*; *Figure 5—figure supplement 4*).

To test the in vivo role of the D-Pax2/Sv sites identified in vitro, we mutated them either individually, or in combinations, in the context of a *dei^ChO-262* reporter transgene encoding for nuclear β-galactosidase (the mutations are shown in *Figure 5—figure supplement 3*). Mutation of the canonical D-Pax2/Sv site 2 reduced the expression level driven by *dei^ChO-262* in cap cells (*Figure 5G and M*). Mutation of D-Pax2/Sv site 1 had no significant effect on its own (*Figure 5F and M*) but led to a complete suppression of *dei^ChO-262* function when combined with a mutation in D-Pax2/Sv site 2 (*Figure 5H and M*). This effect was evident in all types of larval ChOs (*Figure 5H*). These results suggest that D-Pax2/Sv regulates the expression of *dei* in cap cells by binding to two D-Pax2/Sv binding sites within the *dei^ChO-262* enhancer.

We next tested whether the Pros sites identified by EMSA function in vivo to suppress *dei^ChO-262* activity in scolopale cells. Mutating the Pros site 2 resulted in ectopic expression of the reporter gene in the scolopale cells of LCh5 (*Figure 5J and N*). In contrast, disruption of Pros site 1 or site 3 had very small or no detectable effects, respectively, on the expression of *dei^ChO-262* in the scolopale cells of LCh5 (*Figure 5I, K and N*) but induced ectopic expression in scolopale cells of the LCh1 and VChA/B organs (*Figure 5I and K*, arrowheads). In addition, the disruption of Pros site 3 led to elevation in the reporter's level in the LCh5 ligament cells (*Figure 5K*). Simultaneous mutation of all three Pros binding sites in *dei^ChO-262* resulted in intensified ectopic expression in the scolopale cells of all types of larval ChOs (LCh5, LCh1, VChA/B; *Figure 5L and N*). Based on these results we conclude that Pros represses *dei* expression in the scolopale cell by binding to three low-affinity binding sites in *dei^ChO-262* that function additively. It is important to note that the mutations introduced in Pros site 1 also disrupted the low-affinity site for D-Pax2/Sv (site 1, *Figure 5B*) and affected D-Pax2/Sv binding in vitro (*Figure 5—figure supplement 1D*). While the D-Pax2/Sv site 1 is dispensable for *dei* expression in cap cells at the presence of the high-affinity site, it is possible that mutations in Pros site 1 would

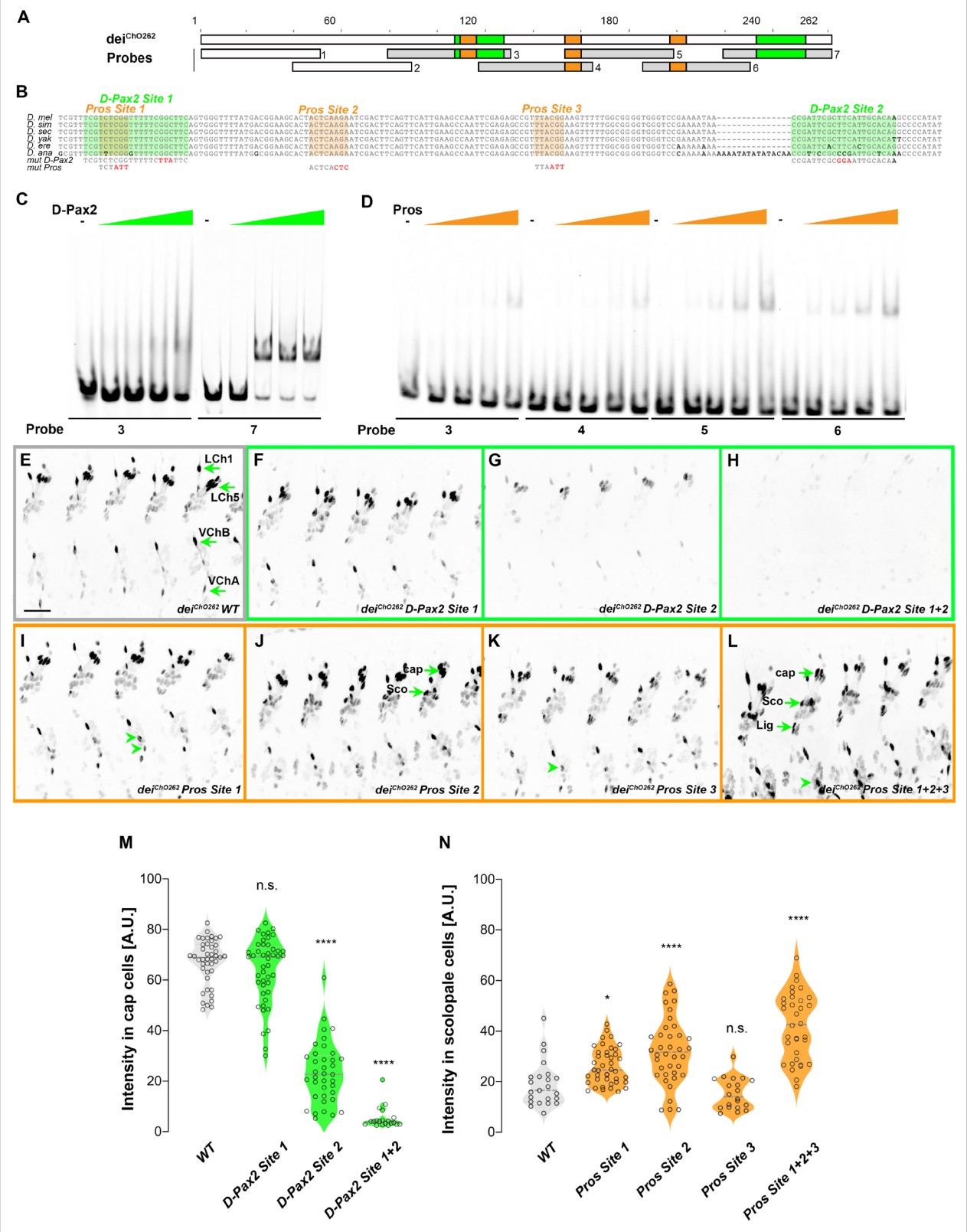

**Figure 5.** *dei^{ChO-262}* contains two binding sites for the activator D-Pax2/Sv and three binding sites for the repressor Pros. (**A**) Schematic representation of the *dei^{ChO-262}* enhancer. Green and orange boxes represent the location of the D-Pax2/Sv and Pros binding sites identified by systematic EMSAs using oligos corresponding to the regions represented by Boxes 1–7. (**B**) Sequence alignment between six *Drosophila* species for the region of the *dei^{ChO-262}* enhancer containing the three D-Pax2/Sv and Pros sites (labeled and highlighted in green and orange, respectively). Dashes indicate gaps in

*Figure 5 continued on next page*

*Figure 5 continued*

the aligned sequence. Mutations of the D-Pax2/Sv and Pros sites used for the in vivo assays are shown at the bottom. (**C**) D-Pax2/Sv binds to two sites in *dei*$^{ChO-262}$, one low affinity site in fragment 3 (*D-Pax2 site 1*) and one high affinity site in fragment 7 (*D-Pax2 site 2*), as demonstrated with EMSA. The full screen is shown in *Figure 5—figure supplement 1*. (**D**) Pros binds to three sites in *dei*$^{ChO-262}$: *Pros site 1* in fragment 3, *Pros site* 2 in an overlapping sequence in fragments 4 and 5, *Pros site 3* in fragment 6, as demonstrated with EMSA. The full screen is shown in *Figure 5—figure supplement 2*. (**E–L**) Expression of wild-type (**E**) and mutated (**F–L**) *dei*$^{ChO-262}$-*lacZ* reporter constructs in abdominal segments A2-A6 of representative stage 16 embryos. The name of the construct is indicated in the bottom of each panel. The green arrows in E point to the cap cells of the various ChO of one abdominal segment: LCh1-lateral ChO1, LCh5-pentascolopidial organ, VChB and VChA are two ventral ChOs. The green arrows in J and L point to cap, scolopale (Sco) and ligament (Lig) cells of LCh5 organs where elevated level of reporter expression is evident. The arrowheads in I and K point to ligament cells of the ventral ChOs. in (**M–N**) Quantification of reporter activity in nuclei of cap (**M**) and scolopale cells (**N**) from LCh5 of segment A2 in embryos carrying the indicated constructs (n = 10 embryos for each genotype). In violin plots, each point represents an individual nucleus, median is shown as dark gray dashed line. Asterisks denote significant difference from wild-type activities, (*) - p < 0.05, (****) – p < 0.0001, n.s. – not significant (Dunnett's multiple comparison test).

The online version of this article includes the following source data and figure supplement(s) for figure 5:

**Figure supplement 1.** Identification of the regions in the *dei*$^{ChO-262}$ enhancer that bind D-Pax2/Sv in vitro.

**Figure supplement 1—source data 1.** I dentification of the regions in the dei$^{ChO-262}$ enhancer that bind D-Pax2 in vitro - the full raw unedited gels.

**Figure supplement 2.** Identification of the regions in the *dei*$^{ChO262}$ enhancer that bind Pros in vitro.

**Figure supplement 2—source data 1.** I dentification of the regions in the dei$^{ChO-262}$ enhancer that bind Pros in vitro - the full raw unedited gels.

**Figure supplement 3.** Sequence of the *dei*$^{ChO-262}$ enhancer and tested mutations.

**Figure supplement 4.** The *dei*$^{ChO-262}$ enhancer contains non-canonical binding sites for Pros and D-Pax2.

**Figure supplement 5.** The D-Pax2 and Pros binding sites regulate the activity of the *dei*$^{ChO-262}$ enhancer in femoral ChOs.

cause a greater effect on the reporter expression in scolopale cells if the D-Pax2/Sv site would not be disrupted simultaneously.

To test whether the identified regulatory interactions apply to additional types of ChOs, we examined the expression pattern driven by the wild-type and mutated *dei*$^{ChO-262}$ enhancers in developing femoral ChOs (*Figure 5—figure supplement 5*). This analysis suggests that the Dei/Pros/D-Pax2 GRN plays a similar role in larval and adult ChOs and that the *dei*$^{ChO-262}$ enhancer encodes pleiotropic transcription factor binding sites (*Preger-Ben Noon et al., 2018*) that integrate the activating and repressing inputs of D-Pax2/Sv and Pros, respectively.

## The *dei*$^{ChO-262}$ enhancer is essential for normal ChO function and larval locomotion

We have previously shown that the cap cell plays a crucial role in propagating muscle-generated mechanicals signals to the sensory neuron (*Hassan et al., 2019*). To test whether *dei* expression in the cap and ligament cells, mediated solely by the *dei*$^{ChO-262}$ enhancer, is essential for the proprioceptive function of the ChO, we analyzed the pattern of locomotion of freely moving *dei*$^{ΔChO}$ larvae and compared it to wild-type larvae and larvae homozygous for the *dei*$^{KO-mCherry}$ null allele (*Hassan et al., 2018*). As shown in *Figure 6*, *wild-type* larvae crawled persistently 94.5% ± 12.4% (n = 27) of the time with a very few changes of direction or head swipes (*Figure 6A, E and H*, and *Videos 1–2*). In contrast, the *dei*$^{KO-mCherry}$ larvae exhibited frequent changes of moving direction and longer pauses, walking on average only 38.9% ± 19.4% (n = 25) of the time. While pausing, the *dei*$^{KO-mCherry}$ larvae swiped their heads extensively (*Figure 6B, F and H*, and *Videos 3–4*). The *dei*$^{ΔChO}$ larvae exhibited locomotion phenotypes that were similar to, though slightly less severe and more variable than, those of the *dei*$^{KO-mCherry}$ larvae (*Figure 6C, G and H*, *Videos 5–6*). On average, the *dei*$^{ΔChO}$ larvae crawled 51.9% ± 24.9% (n = 24) of the time and swiped their heads often; in 15.8% of the time their body angle was higher than 250° or lower than 110°, compared to 9.9% in *dei*$^{KO-mCherry}$ and 1.2% of the time in *wt* larvae (*Figure 6H*). These results demonstrate that the *dei*$^{ChO-262}$ regulatory element is crucial for proper function of the ChO as its deletion resulted in sensory dysfunction and uncoordinated movement.

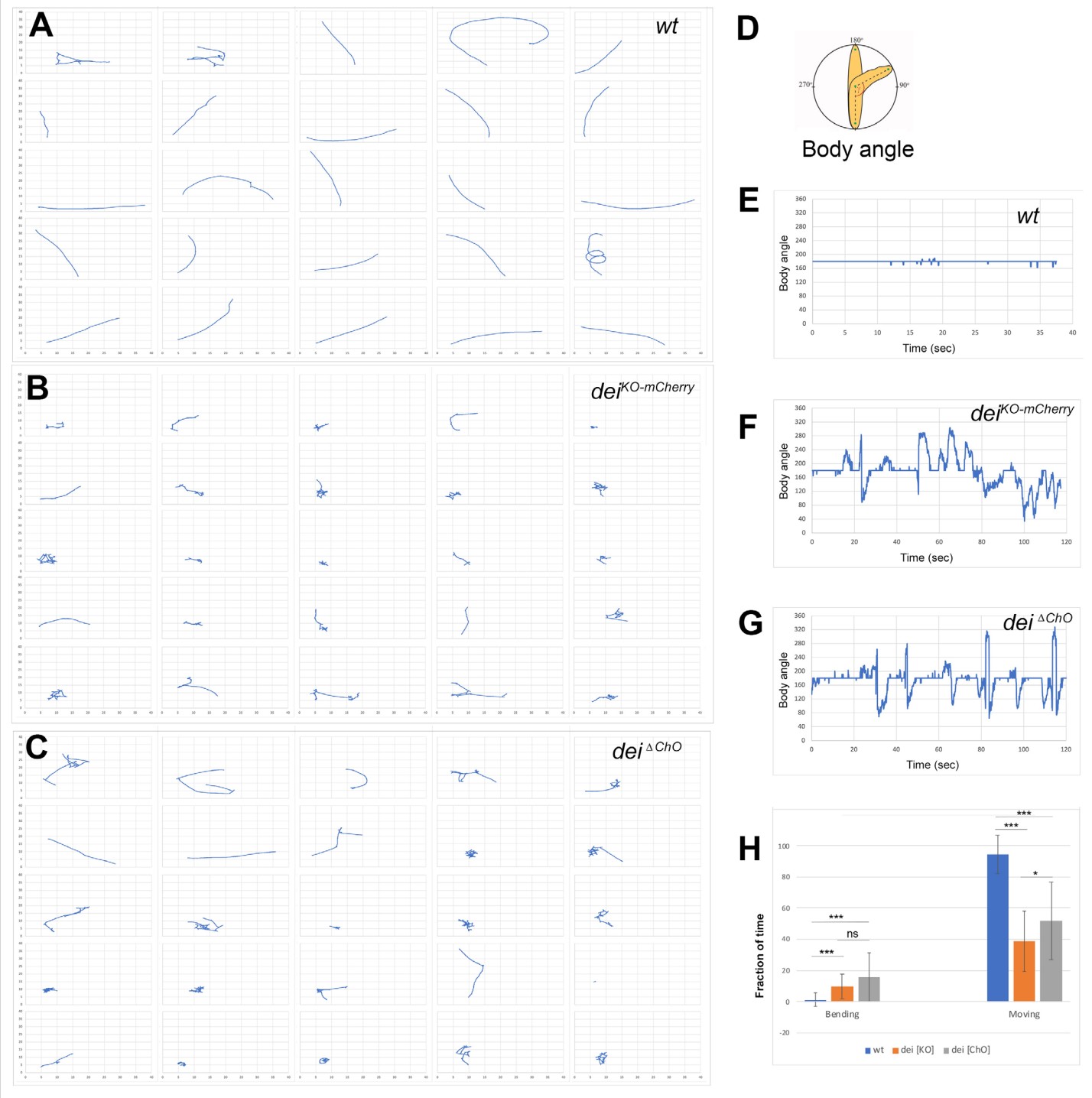

**Figure 6.** The *dei^ChO-262* enhancer is essential for normal larval locomotion. (**A–C**) Crawling trajectories of 25 wild-type (**A**) *dei^KO-mCherry* (**B**) *and dei^ΔChO* (**C**) larvae. Each trajectory is shown in a square that represents 40 × 40 mm area. (**D**) Schematic representation of the body angle, γ, defined as the angle between the head and the body axis. (**E–G**) Representative time evolutions of the body angle of a wild-type (**E**), *dei^KO-mCherry* (**F**) and *dei^ΔChO* (**G**) larvae. The wild-type larva walks persistently, and the body angle stays 180 degrees most of the time (a 40 s interval is shown, after which the larva exited the filmed arena). The *dei^KO-mCherry* and *dei^ΔChO* mutant larvae display frequent changes in the direction of motion and long pauses accompanied by extensive head swiping (120 s intervals are shown). (**H**) A graph showing the average fraction of the time the larvae were crawling (GoPhase) and the fraction of time in which the body was bended more than 70 degrees (the measured angle γ was higher than 250 degrees or lower than 110 degrees). n = 24–27 for all genotypes; error bars represent the standard deviation. *** p < 0.0001, * p < 0.05, ns = non-significant. p Values were calculated using the unpaired two-tailed Mann-Whitney test.

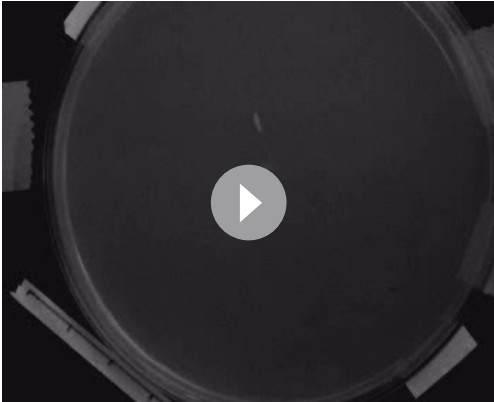

**Video 1.** A video showing the locomotion of a *wt* (*Canton-S*) larva.
https://elifesciences.org/articles/70833/figures#video1

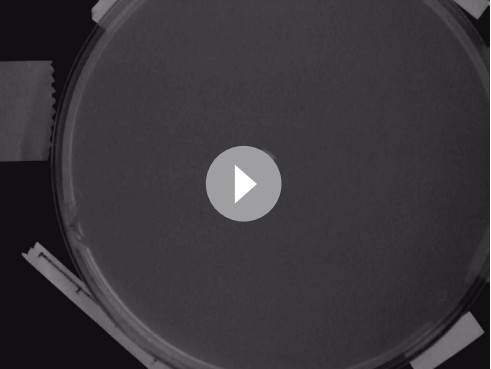

**Video 2.** A video showing the locomotion of a *wt* (*Canton-S*) larva.
https://elifesciences.org/articles/70833/figures#video2

## Discussion

### Opposing activities of D-Pax2/Sv and Pros dictate cap versus scolopale differentiation programs by regulating the *dei* gene

In this work, we identify a small GRN that governs the alternative differentiation programs of two cousins once removed cells within the ChO lineage - the cap cell and the scolopale cell. We show that Pros and D-Pax2/Sv are direct regulators of *dei* that together dictate its expression in the cap cell and its repression in the scolopale cell. Both D-Pax2/Sv and Pros exert their effects on *dei* transcription via a 262 bp chordotonal-specific enhancer (*dei^ChO-262*) in which two D-Pax2/Sv and three Pros binding sites were identified.

Following primary cell fate decisions within the ChO lineage, Pros expression becomes restricted to the scolopale cell (*Doe et al., 1991*; *Vaessin et al., 1991*), whereas D-Pax2/Sv expression becomes restricted to the scolopale and cap cells (*Figure 1A*), similar to its behavior in the external sensory lineages (*Johnson et al., 2011*; *Kavaler et al., 1999*). D-Pax2/Sv activates the expression of *dei* in the cap cell but is unable to do so in the scolopale cell where Pros is co-expressed. If D-Pax2/Sv activity is compromised, the cap cell fails to express *dei* and loses some of its differentiation markers, such as the expression of αTub85E. In contrast, if Pros activity is lost, *dei* is ectopically expressed in the scolopale cell that, as a consequence, acquires some cap cell features including the expression of αTub85E (*Figure 7*). The observed D-Pax2/Sv- and Pros-associated phenotypes do not reflect genuine cell fate transformations, suggesting that D-Pax2/Sv and Pros do not affect primary cell fate decisions within the ChO lineage. Rather, the observed phenotypes reflect a failure of the cap and scolopale cells to follow the cell type-specific differentiation programs responsible for bringing about their

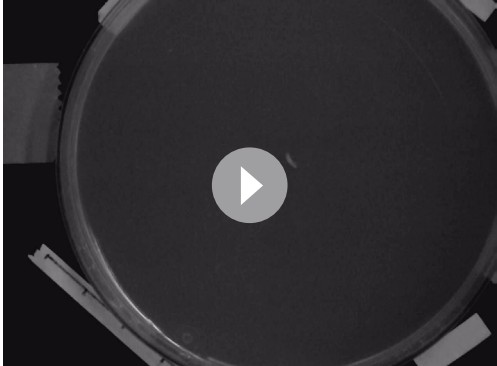

**Video 3.** A video showing the locomotion of a *dei*^KO-mCherry larva.
https://elifesciences.org/articles/70833/figures#video3

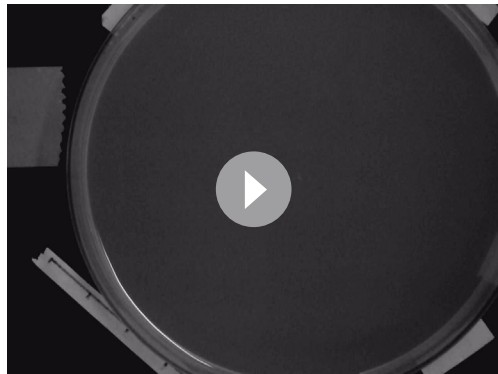

**Video 4.** A video showing the locomotion of a *dei*^KO-mCherry larva.
https://elifesciences.org/articles/70833/figures#video4

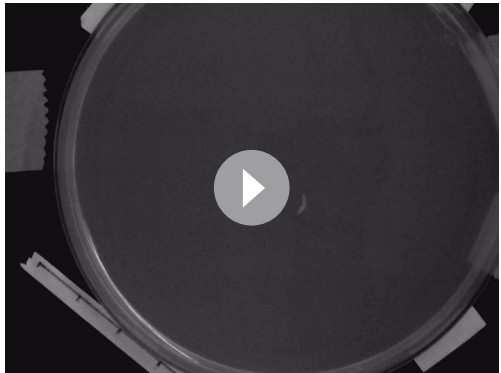

**Video 5.** A video showing the locomotion of a *dei*^ΔChO^ larva.
https://elifesciences.org/articles/70833/figures#video5

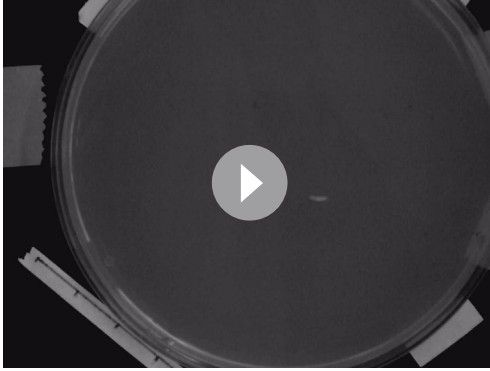

**Video 6.** A video showing the locomotion of a dei^ΔChO^ larva.
https://elifesciences.org/articles/70833/figures#video6

characteristic cellular phenotypes. The D-Pax2/Sv-deficient cap cells fail to express unique differentiation markers (such as αTub85E) and are therefore hardly detectable. It is also possible that the Sv/Pax2-deficient cap cells fail to survive. Thus, we cannot exclude the possibility that some of the findings reflect more upstream roles of Sv/D-Pax2 in the specification of cap-cell identity.

The switch between the differentiation programs of cap and scolopale identities cannot be simply explained by the nature of asymmetric cell divisions within the ChO lineage. The effects on the ChO lineage of major regulators of asymmetric cell division, such as Notch and Numb, and the expression pattern of cell differentiation determinants such as Pros and D-Pax2/Sv, were mainly postulated based on knowledge gained by analyzing external sensory lineages (*Kavaler et al., 1999*; *Lai and Orgogozo, 2004*; *Manning and Doe, 1999*; *Reddy and Rodrigues, 1999*). According to the similarity between the lineages, the cap cell parallels the Notch-non-responsive hair (trichogen) cell, whereas the scolopale parallels the Notch- responder sheath (thecogen) cell (based on *Rebeiz et al., 2011*). Thus, D-Pax2/Sv is expressed in one Notch responder and one non-responder cells in the lineage. The presence of Pros in the Notch-responder cell represses the cap-promoting activity of D-Pax2/Sv. Somewhat similar cousin-cousin cell transformation was found in external sensory organs in the adult where mutations in *hamlet* transform the sheath cell into a hair cell (parallel to scolopale-to-cap transformation) (*Moore et al., 2004*). Ectopic expression of *hamlet* induced *pros* expression and repressed the hair shaft-promoting activity of D-Pax2/Sv.

In the adult external sensory lineage, Pros was shown to be important for the specification of the pIIb precursor, which gives rise to the neuron and sheath cell (the scolopale counterpart). However, the absence of Pros from the pIIa precursor, which gives rise to the hair and socket cells (the cap and cap-attachment cells counterparts) was even more critical for proper development of this branch of the lineage (*Manning and Doe, 1999*). This phenomenon is somewhat conserved in the larval ChO. While Pros is required for proper differentiation of the scolopale cell, its absence from the cap cell is critical for adopting the correct differentiation programs within the lineage.

Opposing effects of D-Pax2/Sv and Pros activities on cell differentiation have been also identified in the regulation of neuronal versus non-neuronal cell fate decisions in the developing eye, where they play a role in modulating the Notch and Ras signaling pathway (*Charlton-Perkins et al., 2011*). Interestingly, in the R7 equivalence group Pros and D-Pax2/Sv can only alter the cell-type-specific differentiation program of cells that already express the other gene (*Charlton-Perkins et al., 2011*). Similarly, in the ChO lineage, ectopic expression of Pros in the cap and ligament cells transforms the D-Pax2/Sv-positive cap cell toward a scolopale cell identity but does not affect the D-Pax2/Sv-negative ligament cell in a similar fashion, even though the ectopic expression of Pros does repress the transcription of *dei* in both cell types. Additionally, a loss of Pros activity in the scolopale cell can transform the identity of this cell toward a cap cell identity only in the presence of D-Pax2/Sv.

We have shown that the opposing influences of Pros and D-Pax2/Sv on *dei* expression is integrated by the *dei*^ChO-262^ enhancer in both larval and adult ChO lineages. To the best of our knowledge, this is the first example of an enhancer that responds to these opposing signals to dictate cell-specific

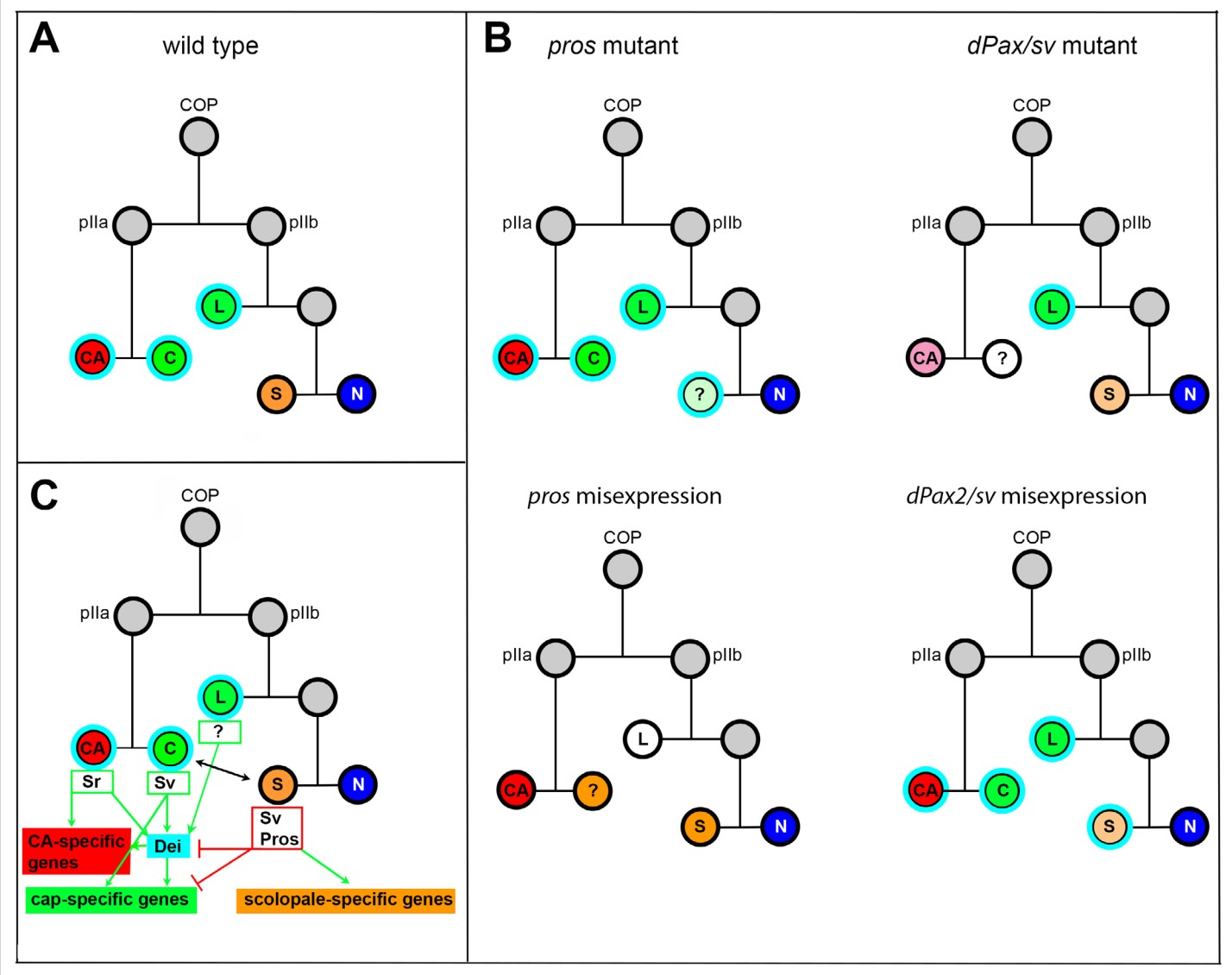

**Figure 7.** Summary of the relations between Sv, Pros and Dei in the ChO lineage and their effect on ChO development. (**A**) A *wt* ChO lineage. The cap (C) and ligament (L) cells are depicted in green, the scolopale cell (S) is depicted in orange and the neuron (N) is depicted in blue. Cells that expressed Dei are circled in light blue. (**B**) The loss of pros leads to upregulation of Dei in the scolopale cell and to failure of scolopale cell differentiation. In contrast, the loss of Sv leads to loss of Dei expression from the cap cell and failure of cap cell differentiation. The Sv-deficient scolopale cells are also abnormal. The CA cells which depend on the cap cell for their development/maintenence also appear abnormal is *sv* mutants. Misexpression of Pros leads to repression of Dei in the cap and ligament cells, preventing their normal differentiation. The Pros-expressing cap cells adopt some scolopale-specific features. In contrast, over-expression of Sv leads to ectopic expression of Dei. Due to the presence of Pros, the level of expression of *dei* in the scolopale cell is restricted. (**C**) A schematic summary showing the relations between Pros, Sv and Dei and their relations to cell-type-specific differentiation programs. In the CA cells, *dei* is activated by Sr via the *dei^attachment* enhancer. Both Sr and Dei are required there for the activation of CA-specific genes. In the cap cell *dei* is activated by Sv via the *dei^ChO-262* enhancer. Sv is required for activating cap-specific genes in both Dei-dependent and independent ways. In the scolopale cells, *dei* is repressed by Pros via the *dei^ChO-262* enhancer. Pros is required in addition for activating scolopale-specific genes. Dei is also expressed in the ligament cells and is required for their correct differentiation. The regulators of *dei* in the ligament cell are yet to be identified.

differentiation programs in a sensory lineage. While the identified enhancer is ChO-specific, it is plausible that other enhancers of sensory organ lineage-specific genes encode coupled Pros and D-Pax2/Sv binding sites. The expression of the *dei* gene in other (non-ChO) organs is regulated via different enhancers (as described in *Nachman et al., 2015*). Some of these enhancers are responsible for regulating *dei* expression in tissues where Pros and Pax2 play opposing roles, such as the eye and wing margin ES organs (the *dei^wing+eye* enhancer; *Nachman et al., 2015*). It is beyond the scope of this work,

but in the future, it will be interesting to decipher whether these enhancers also serve as molecular platforms for integrating opposing effects of Pax2 and Pros.

## The *dei*<sup>ChO-262</sup> enhancer encodes non-canonical binding sites for Pros and D-Pax2/Sv

We have identified two D-Pax2/Sv and three Pros binding sites in the *dei*<sup>ChO-262</sup> enhancer. Apart from D-Pax2/Sv site 2, none of these sites match the published binding motifs for D-Pax2/Sv or Pros. These results agree with recent studies that showed that many transcription factors function in vivo through low-affinity (*Crocker et al., 2016*; *Crocker et al., 2015*) or suboptimal (*Farley et al., 2015*) binding sites that differ from their predicted binding motifs. It was suggested that low-affinity binding sites provide specificity for individual transcription factors belonging to large paralogous families, such as the homeodomain family of transcription factors, that share similar DNA-binding preferences (*Crocker et al., 2016*; *Crocker et al., 2015*; *Kribelbauer et al., 2019*). To compensate for their weak binding capabilities, low-affinity binding sites are often organized in homotypic clusters that can increase the cumulative binding affinity of an enhancer (*Crocker et al., 2016*; *Crocker et al., 2015*; *Kribelbauer et al., 2019*). Our findings, that the homeodomain transcription factor Pros functions through a cluster of low-affinity binding sites in *dei*<sup>ChO-262</sup>, may represent another example for the suggested tradeoff between transcription factor binding affinity and specificity (*Crocker et al., 2016*; *Crocker et al., 2015*).

We do not know how Pros opposes the effect of D-Pax2/Sv in the context of *dei*<sup>ChO-262</sup> to inhibit the expression of *dei* in scolopale cells. Our results suggest that the inhibitory effect of Pros is not mediated through binding competition with D-Pax2/Sv at the D-Pax2/Sv high-affinity site (site 2), since this site does not overlap with a Pros-binding site (*Figure 5B*). The D-Pax2/Sv low-affinity site does overlap with a Pros binding site and mutations in the Pros binding site affect D-Pax2/Sv binding in vitro (see *Figure 5—figure supplement 1D*), however, while being important for robust *dei* expression, this site is dispensable in the presence of the high-affinity site. It is possible that binding of Pros to the *dei*<sup>ChO-262</sup> enhancer targets this sequence to a repressed heterochromatin domain as was recently shown for other Pros target genes in differentiating neurons (*Liu et al., 2020*).

How is *dei* regulated in other ChO cell types? *dei* is expressed in four out of six cell types comprising the ChO: the cap-attachment and ligament-attachment cells, in which *dei* transcription is activated by Sr via the *dei*<sup>attachment</sup> regulatory module (*Nachman et al., 2015*), and the cap and ligament cells in which the expression of *dei* is regulated via the *dei*<sup>ChO-262</sup> enhancer. We now show that D-Pax2/Sv activates *dei* transcription in the cap cell, and that Pros inhibits its expression in the scolopale cell. The identity of the positive regulator/s of *dei* in the ligament cell, whose cell-fate is determined by the glial identity genes *gcm* and *repo* (*Campbell et al., 1994*; *Halter et al., 1995*; *Jones et al., 1995*), and the identity of the negative regulator/s of *dei* in the neuron remains unknown. Interestingly, the expression of *dei* was found to be altered in response to ectopic expression of *gcm* in the embryonic nervous system; its expression was upregulated at embryonic stage 11, but was repressed in embryonic stages 15–16 (*Egger et al., 2002*). This observation points to GCM as a potential regulator of *dei* expression in the ligament cells. Another interesting candidate for repressing *dei* in the sensory neuron is the transcriptional repressor Lola. Lola has been identified as a putative direct regulator of *dei* in the Y1H screen and was shown to be required in post-mitotic neurons in the brain for preserving a fully differentiated state of the cells (*Southall et al., 2014*). The possible involvement of Gcm and Lola in the regulation of *dei* awaits further studies. The observed upregulation of the *dei*<sup>ChO-262</sup> reporter in the ligament cells of embryos with mutated Pros-binding sites may reflect an early role of Pros in the pIIb precursor before its restriction to the scolopale cell, which prevents *dei* expression in the ligament cell.

## The *dei*<sup>ChO-262</sup>-driven *dei* expression is critical for organ functionality

Although the loss of *dei* in the genetic/cellular milieu of the ligament cell (unlike the cap cell), even when accompanied by ectopic expression of Pros, is not sufficient for transforming ligament cell properties towards those of scolopale cells, we know that the expression of *dei* in the ligament cell is critical for its proper development. Ligament-specific knockdown of *dei* leads to failure of the ligament cells to acquire the right mechanical properties and leads to their dramatic over-elongation (*Hassan et al., 2018*). By analysing the locomotion phenotypes of larvae homozygous for a *dei* null allele and

the newly generated cap and ligament-specific $dei^{\Delta ChO}$ allele, we could show that the expression of *dei* in the cap and ligament cells is crucial for normal locomotion. Thus, we conclude that the correct expression of *dei* within the ChO is critical for organ functionality. Surprisingly, the gross morphology of LCh5 of $dei^{\Delta ChO}$ larvae appears normal (Data not shown). Yet, in a way that remains to be eluci-dated, the Dei-deficient cap and ligament cells fail to correctly transmit the cuticle deformations to the sensory neuron, most likely due to changes in their mechanical properties.

# Materials and methods

**Key resources table**

| Reagent type (species) or resource | Designation | Source or reference | Identifiers | Additional information |
|---|---|---|---|---|
| Gene (*Drosophila melanogaster*) | *dei/tx* | FlyBase | CG5441, FBgn0263118 | |
| Gene (*Drosophila melanogaster*) | *pros* | FlyBase | CG17228, FBgn0004595 | |
| Gene (*D. melanogaster*) | *sv/D-Pax2* | FlyBase | CG11049, FBgn0005561 | |
| Genetic reagent (*D. melanogaster*) | $sv^6$/act-GFP | *Kavaler et al., 1999* | N/A | |
| Genetic reagent (*D. melanogaster*) | Dpax2$^{D1}$-GFP | *Johnson et al., 2011* | N/A | |
| Genetic reagent (*D. melanogaster*) | pros$^{17}$/TM6B, Tb$^1$ | Bloomington *Drosophila* Stock Center | BDSC:5458 | |
| Genetic reagent (*D. melanogaster*) | $dei^{ChO-1353}$-GFP,$dei^{attachment}$-RFP;en-Gal4 | *Halachmi et al., 2016* | N/A | |
| Genetic reagent (*D. melanogaster*) | $dei^{KO-mCherry}$ | *Hassan et al., 2018* | N/A | |
| Genetic reagent (*D. melanogaster*) | en-Gal4 | *Brand and Perrimon, 1993* | | |
| Genetic reagent (*D. melanogaster*) | P{UAS-3xFLAG-pros.S}14 c, y1 w*; Pin$^1$/CyO | Bloomington *Drosophila* Stock Center | BDSC:32245 | |
| Genetic reagent (*D. melanogaster*) | UAS-sv-RNAi | Vienna *Drosophila* Resource Center | VDRC:107343 | |
| Genetic reagent (*D. melanogaster*) | UAS-sv | *Kavaler et al., 1999* | N/A | |
| Genetic reagent (*D. melanogaster*) | UAS-D-α-Catenin-GFP | *Oda and Tsukita, 1999* | N/A | |
| Genetic reagent (*D. melanogaster*) | $dei^{attachment}$-lacZ | *Nachman et al., 2015* | N/A | |
| Genetic reagent (*D. melanogaster*) | $dei^{ChO-1353}$-lacZ | *Nachman et al., 2015* | N/A | |
| Genetic reagent (*D. melanogaster*) | $dei^{ChO-1353}$-GFP | *Halachmi et al., 2016* | N/A | |
| Genetic reagent (*D. melanogaster*) | $dei^{ChO-389}$-lacZ | *Nachman et al., 2015* | N/A | |
| Genetic reagent (*D. melanogaster*) | $dei^{ChO}$-Gal4 | This study | N/A | |
| Genetic reagent (*D. melanogaster*) | $dei^{ChO-262}$-lacZ (in pH-Pelican) | This study | N/A | P element transgenesis. Available on 1st, 2nd, and 3rd chromosomes |
| Genetic reagent (*D. melanogaster*) | $dei^{ChO-262}$-placZ-wildtype (in placZattB) | This study | N/A | Inserted at attP2 |

*Continued on next page*

*Continued*

| Reagent type (species) or resource | Designation | Source or reference | Identifiers | Additional information |
|---|---|---|---|---|
| Genetic reagent (*D. melanogaster*) | *dei^ChO-262-placZ-Pros-site 1* (in *placZattB*) | This study | N/A | Inserted at *attP2* |
| Genetic reagent (*D. melanogaster*) | *dei^ChO-262-placZ-Pros-site 2* (in *placZattB*) | This study | N/A | Inserted at *attP2* |
| Genetic reagent (*D. melanogaster*) | *dei^ChO-262-placZ-Pros-site 3* (in *placZattB*) | This study | N/A | Inserted at *attP2* |
| Genetic reagent (*D. melanogaster*) | *dei^ChO-262-placZ-Pros-site 1 + 2 + 3* (in *placZattB*) | This study | N/A | Inserted at *attP2* |
| Genetic reagent (*D. melanogaster*) | *dei^ChO-262-placZ-Pax2-site 1* (in *placZattB*) | This study | N/A | Inserted at *attP2* |
| Genetic reagent (*D. melanogaster*) | *dei^ChO-262-placZattB Pax2-site 2* | This study | N/A | Inserted at *attP2* |
| Genetic reagent (*D. melanogaster*) | *dei^ChO-262-placZ-Pax2-site 1 + 2* (in *placZattB*) | This study | N/A | Inserted at *attP2* |
| Genetic reagent (*D. melanogaster*) | *dei^ΔChO/(TM6)* | This study | N/A | |
| Genetic reagent (*D. melanogaster*) | *M{nos-Cas9.P}ZH-2A* | Bloomington *Drosophila* Stock Center | BDSC:54591 | |
| Antibody | Anti Sv/D-Pax2 (rabbit polyclonal) | *Johnson et al., 2011* | N/A | (1:10,000) |
| Antibody | Anti-α85E-Tubulin (rabbit polyclonal) | *Klein et al., 2010* | N/A | (1:200) |
| Antibody | Anti α85E-Tubulin (mouse monoclonal) | *Nachman et al., 2015* | N/A | (1:20) |
| Antibody | Anti-Dei (rabbit polyclonal) | *Egoz-Matia et al., 2011* | N/A | (1:50) |
| Antibody | Anti-Spalt (rabbit polyclonal) | *Halachmi et al., 2007* | N/A | (1:500) |
| Antibody | Anti-NRG (rat polyclonal) | *Banerjee et al., 2006* | N/A | (1:1000) |
| Antibody | Anti-βGal (mouse monoclonal) | Promega | Z3781 | (1:1000) |
| Antibody | Anti-Pros (mouse monoclonal) | The Developmental Studies Hybridoma Bank | MR1A | (1:20) |
| Antibody | Anti Futsch (mouse monoclonal) | The Developmental Studies Hybridoma Bank | 22C10 | (1:20) |
| Antibody | Anti-ELAV (rat monoclonal) | The Developmental Studies Hybridoma Bank | 7E8A10 | (1:50) |
| Antibody | Anti-Eys (mouse monoclonal) | The Developmental Studies Hybridoma Bank | 21A6 | (1:20) |
| Antibody | Anti-Repo (mouse monoclonal) | The Developmental Studies Hybridoma Bank | 8D12 | (1:10) |
| Antibody | Anti Crb (mouse monoclonal) | The Developmental Studies Hybridoma Bank | Cq4 | (1:10) |
| Antibody | Anti-Cpo (rabbit polyclonal) | *Bellen et al., 1992* | N/A | (1:5000) |
| Antibody | Anti-Sr (Chicken polyclonal) | This study | N/A | (1:20) |
| Antibody | Cy3-conjugated goat anti-mouse | Jackson Laboratories, Bar-Harbor, Maine, USA | 115-165-166 | (1:100) |
| Antibody | Cy3-conjugated goat anti-rabbit | Jackson Laboratories, Bar-Harbor, Maine, USA | 115-165-144 | (1:100) |

*Continued on next page*

Continued

| Reagent type (species) or resource | Designation | Source or reference | Identifiers | Additional information |
|---|---|---|---|---|
| Antibody | Cy2-conjugated goat anti-rabbit | Jackson Laboratories, Bar-Harbor, Maine, USA | 111-225-144 | (1:100) |
| Antibody | Alexa Fluor-647-conjugated goat anti-mouse | Jackson Laboratories, Bar-Harbor, Maine, USA | 115-605-166 | (1:100) |
| Antibody | Alexa Fluor-647-conjugated goat anti-rabbit | Jackson Laboratories, Bar-Harbor, Maine, USA | 115-605-144 | (1:100) |
| Antibody | Cy3-conjugated donkey anti-chicken IgY | Jackson Laboratories, Bar-Harbor, Maine, USA | 703-165-155 | (1:100) |
| Antibody | Alexa Fluor-647-conjugated Donkey anti-Rat | Jackson Laboratories, Bar-Harbor, Maine, USA | 712-605-153 | (1:100) |
| Commercial assay or kit | qScript cDNA Synthesis kit | Quanta BIOSCIENCES | 95047–100 | |
| Commercial assay or kit | PrimeSTAR Max DNA polymerase | TAKARA | R045A | |
| Chemical compound, drug | DAKO fluorescence mounting medium | Agilent Technologies, Santa Clara, CA, USA | S3023 | |
| Chemical compound, drug | TRI Reagent | Sigma | T9424 | |
| Software, algorithm | ImajeJ | National Institute of Health | SCR_003070 | |
| Software, algorithm | FIMTrack tracking software | *Risse et al., 2017* | https://www.uni-muenster.de/FRIA/en/FIM/ | |
| Software, algorithm | MATLAB | Mathworks | SCR_001622 | |
| Software, algorithm | Imaris | Bitplane | SCR_007370 | |

## Fly strains

The following mutant and reporter alleles of *sv* and *pros* were used: *sv⁶/act-GFP* (*Kavaler et al., 1999*), *Dpax2^D1-GFP* (*Johnson et al., 2011*), *pros^17/TM6B, Tb^1* (BDSC:5458). The following Gal4 drivers and UAS strains were used: *dei^ChO-1353-GFP,dei^attachment-RFP;en-Gal4* (*Halachmi et al., 2016*), *dei^KO-mCherry* (*Hassan et al., 2018*), *en-Gal4* (*Brand and Perrimon, 1993*), P{UAS-3xFLAG-pros.S}14 c, y1 w*; Pin^1/CyO (BDSC:32245). UAS-sv-RNAi (VDRC:107343), *UAS-sv* (*Kavaler et al., 1999*), UAS-D-α-Catenin-GFP (*Oda and Tsukita, 1999*).

The *dei^ChO-Gal4* driver was constructed by cloning the *dei^ChO-1353* regulatory module described in *Nachman et al., 2015* into the *pChs-gal4* vector which was then used for the generation of transgenic fly strains (insertions are available on the X, 2nd and 3rd chromosome). The dei^ChO–262-lacZ strain was generated as previously described in *Nachman et al., 2015*. For mutational analysis of the putative binding sites, wild-type and mutated *dei^ChO-262* fragments were synthesized by GenScript (USA) and cloned into the reporter constructs *placZattB*. Plasmids were integrated into the *attP2* landing site by BestGene Inc (Chino Hills CA, USA). (*Supplementary file 2* lists all the transgenes used in this study, vectors used, landing sites and the sequence of the inserted mutations). The *dei^ΔChO* allele was generated by GenetiVision (Houston TX, USA) via multiplex targeting with two sgRNAs: 5'GCAC TTGTTTGCGTTTACATTAC3' and 5'GGCGAGAAGTATTCCCTGCG3'; creating a defined deletion of 307 bps spanning the *dei^ChO-262* fragment. The presence of the desired deletion was verified by sequencing. To verify that this deletion does not affect splicing or other structural features of the transcript, cDNA was synthesized and sequenced from the homozygous *dei^ΔChO* flies and control flies (the M{nos-Cas9.P}ZH-2A strain to which the injection was done). Total RNA was isolated from 10 adult flies using TRI Reagent (Sigma #T9424) according to the protocol described by Green et al. (Cold Spring Harb Protoc; doi:10.1101/pdb.prot101675). One mg of total RNA was used to generate cDNA using qScript cDNA Synthesis kit, according to the manufacturer protocol (Quanta BIOSCIENCE). PCR amplification was performed on 100 ng of cDNA using PrimeSTAR Max DNA polymerase (TAKARA #R045A), a forward primer from exon 1: TGCCAAATTTATGCATGAGC and reverse primer from exon 2: GCTTCTGTCGCAGGGAATAC.

## Embryo staining and image analysis

Immunostaining of whole-mount embryos was performed using standard techniques. The following primary antibodies used in this study were: Rabbit anti Sv/D-Pax2 (1:10,000) (*Johnson et al., 2011*), Rabbit anti α85E-Tubulin (1:200) (*Klein et al., 2010*), Mouse anti α85E-Tubulin (1:20) (*Nachman et al., 2015*), Rabbit anti Dei (1:50) (*Egoz-Matia et al., 2011*), Rabbit anti Spalt (1:500) (*Halachmi et al., 2007*), Rat anti NRG (1:1000) (*Banerjee et al., 2006*), anti-Cpo (1:5000) (*Bellen et al., 1992*), and Mouse anti-βGal (1:1000, Promega). The following antibodies were obtained from the Developmental Studies Hybridoma Bank: Mouse anti Pros (MR1A, 1:20), Mouse anti Futsch (22C10, 1:20), Rat anti-ELAV (7E8A10, 1:50), Mouse anti-Eys (21A6, 1:20), Mouse anti-Repo (8D12, 1:10), Mouse anti Crb (Cq4, 1:10). Chicken anti-Sr (1:20) was made against amino acids 707–1180 of the Sr protein fused to GST in the pGEX-KG expression vector. The ~80 kDa fusion protein was purified on Glutathione-agarose beads followed by elution with reduced glutathione. Antibodies against the GST-Sr fusion protein were produced in Chickens by Dr. Enav Bar-Shira (Department of Animal Sciences, Robert H. Smith Faculty of Agriculture Food and Environment, The Hebrew University of Jerusalem, Rehovot, Israel). The IgY antibody fraction was isolated from the egg yolk and cleaned on Glutathione-agarose beads to reduce background of anti-GST antibodies. Secondary antibodies for fluorescent staining were Cy3, Cy2, Cy5 or Alexa Fluor-647-conjugated anti-Mouse/Rabbit/Rat/Chicken/Guinea pig (1:100, Jackson Laboratories, Bar-Harbor, Maine, USA). Stained embryos were mounted in DAKO fluorescence mounting medium (Agilent Technologies, Santa Clara, CA, USA) and viewed using confocal microscopy (Axioskop and LSM 510, Zeiss).

For the analysis of reporter gene expression, images were analyzed using ImageJ software (http://rsb.info.nih.gov/ij/) as previously described (*Preger-Ben Noon et al., 2018*). Briefly, maximum projections of confocal stacks were assembled, and background was subtracted using a 50-pixel rolling-ball radius. Then, we manually segmented visible nuclei of cap and scolopale cells from LCh5 of abdominal segments A2 and measured the fluorescence mean intensities of each nucleus. Statistical analyses and graphing were performed using GraphPad Prism version 8, GraphPad Software, La Jolla California USA, https://www.graphpad.com/.

## X-Gal staining

Staining was done on staged pupae collected 40 hr after pupal formation. Pupae were removed from the pupal case and fixed at room temprature for 15 min in 4% formaldehyde in PBS. Following two washes in PBT (PBS + 0.1% Triton X-100), the pupae were incubated for five minutes in X-Gal staining buffer (without X-Gal) (5 mM $_{K4}[Fe^{+2}(CN)_6]$, 5 mM $_{K3}[Fe^{+2}(CN)_6]$) in PBT and then incubated in staining buffer containing 1 mg/ml X-Gal for 4 hr at 37°C.

## Yeast one-hybrid analysis

Yeast one-hybrid screening was performed by Hybrigenics Services, S.A.S., Evry, France (http://www.hybrigenics-services.com). The sequence of *dei^{ChO-389}* was PCR-amplified and cloned into the integrative vector pB301 (pAbAi, Clontech Laboratories, Inc). The construct was checked by sequencing the entire insert and transformed into the YM955 yeast strain to integrate the DNA bait into the yeast genome. Screening was performed against a random-primed *Drosophila* Whole Embryo cDNA library constructed into *pP6* that derives from the original *pGADGH* (*Bartel, 1993*) plasmid. 124 million clones (12-fold the complexity of the library) were screened using a mating approach with YHGX13 (Y187 ade2-101::loxP-kanMX-loxP, matα) and the *dei*-containing yeast (mata) strain as previously described (*Fromont-Racine et al., 1997*). 146 His + colonies were selected on a medium lacking uracil, leucine and supplemented with 400 ng/ml Aureobasidin A. The prey fragments of the positive clones were amplified by PCR and sequenced at their 5' and 3' junctions. The resulting sequences were used to identify the corresponding interacting proteins in the GenBank database (NCBI) using a fully automated procedure. A confidence score (PBS, for Predicted Biological Score) was attributed to each interaction as previously described (*Formstecher et al., 2005*; *Rain et al., 2001*; *Wojcik et al., 2002*).

## Motif search analysis

The D-Pax2/Sv site 2 was predicted by JASPAR (*Mathelier et al., 2016*). The PWM for Pros was generated using the sequences selected in TDA by *Hassan et al., 1997* and the MEME suite (*Bailey*

*et al., 2015*). The MEME suite was used to search for the Pros motif in *dei*[ChO-262] and to compare the experimentally identified binding sites to the D-Pax2/Sv and Pros PWMs.

## Protein purification

*D-Pax2-HD* and *Pros-S-HD* expression plasmids were a kind gift from Brian Gebelein and Tiffany Cook (*Cook et al., 2003*; *Li-Kroeger et al., 2012*). Proteins were purified from *E. coli* (BL21) as described previously (*Uhl et al., 2010*) with the following modifications. Protein expression was induced at 37 °C using 0.1 mM IPTG for 4 hr (Pros-S/L-HD) or 0.4 mM IPTG for 4 hr (D-Pax2-HD).

Pros-S-HD: After induction, bacterial pellet was resuspended in PBS supplemented with complete protease inhibitor mix (Roche) and lysed on ice using sonication (10 cycles of 30 s on/off). GST-tagged Pros-S-HD in soluble fraction was purified using Glutathione-Agarose beads (Sigma). The bound proteins were eluted in elution buffer (50 mM Tris, pH 8, 10 mM reduced Glutathione).

D-Pax2-HD: The bacterial pellet was resuspended in 20 mM Tris, pH 7.5 supplemented with protease inhibitor. His-tagged proteins in soluble fraction were purified using cOmplete His-Tag Purification Columns (Roche). The columns were washed with 10 column volumes of wash buffer 1 (20 mM Tris, pH7.5, 300 mM NaCl, 50 mM $NaH_2PO_4$ pH 8.0), 2 column volumes of wash buffer 2 (20 mM Tris, pH7.5, 300 mM NaCl, 5 mM DTT, 10 mM Imidazole), and bound proteins were eluted in the same buffer supplemented with 250 mM Imidazole.

All samples were dialyzed against 500 ml of dialysis buffer (20 mM HEPES, pH 7.9, 200 mM NaCl, 10% Glycerol, 2 mM MgCl) for 18 hr at 4 °C. Protein concentrations were measured with NanoDrop and confirmed by SDS-PAGE and Coomassie blue analysis.

## Electromobility shift assay (EMSA)

DNA probes were generated by annealing 5' IRDye700 labeled forward oligonucleotides with unlabeled reverse oligonucleotides (Integrated DNA Technologies) to a final concentration of 5 µM in PNK buffer (New England Biolabs). One hundred femtomoles of labeled IRDye700 probes were used in a 20 µl binding reaction containing 10 mM Tris, pH 7.5; 50 mM NaCl; 1 mM MgCl2; 4% glycerol; 0.5 mM DTT; 0.5 mM EDTA; 50 µg/ml poly(dI–dC); 200 µg/ml of BSA and purified proteins (see *Supplementary file 3* for amount of each protein used). The binding reactions were incubated at room temperature for 30 min, and run on a native 4% polyacrylamide gel for 1.5 hr at 180 V. For competition assays, the appropriate amount of cold competitor was added with the IRDye700-labeled probe prior to the incubation. The polyacrylamide gel cassettes were imaged using an Odyssey Infrared Imaging System and image analysis was performed using ImageQuant 5.1 software. All experiments were performed at least three times.

## Locomotion assays

Larvae used in the locomotion assays were collected from 8 to 12 hr egg collections that aged at 24°C until reaching the wandering 3rd instar stage (115–140 hr). 25–30 larvae of each genotype were individually transferred to a fresh 2% agar 10 cm plate, prewarmed to 24 °C. Larvae were let to adjust for 30 s prior to 2-min recording at a rate of 30 frames per second. The wild-type larvae often exited the filmed arena before the completion of the full 2 min recording time. Larval locomotion was recorded using a Dino-Lite digital microscope placed above the plate. We used VideoPad software to convert Dino-Lite files into Tiff files. ImageJ and FIMTrack (*Risse et al., 2017*) tracking software were used for following larval (center of mass) movements and body angle.

## Acknowledgements

We wish to thank Joshua Kavaler, Tiffany Cook, Markus Noll, Anna Jazwinska, Manzur Bhat, the Bloomington Stock Center and the Developmental Studies Hybridoma Bank at the University of Iowa for sending us antibodies and fly strains. We are very grateful to Enav Bar-Shira for her kind help in generating the chicken anti-Sr antibodies and to Abeer Hassan for helping us with the analysis of the locomotion assays' data. This work was supported by a grant (No. 674/17) to AS from The Israel Science Foundation.

# Additional information

## Funding

| Funder | Grant reference number | Author |
|---|---|---|
| Israel Science Foundation | Adi Salzberg 674/17 | Adi Salzberg |

The funders had no role in study design, data collection and interpretation, or the decision to submit the work for publication.

## Author contributions

Adel Avetisyan, Conceptualization, Data curation, Investigation, Methodology, Project administration; Yael Glatt, Maya Cohen, Yael Timerman, Naomi Halachmi, Data curation, Investigation; Nitay Aspis, Atalya Nachman, Investigation; Ella Preger-Ben Noon, Data curation, Investigation, Methodology, Writing – original draft, Writing – review and editing; Adi Salzberg, Conceptualization, Data curation, Funding acquisition, Project administration, Resources, Supervision, Writing – original draft, Writing – review and editing

## Author ORCIDs

Adel Avetisyan (iD) http://orcid.org/0000-0002-1196-8485
Yael Timerman (iD) http://orcid.org/0000-0001-5509-0275
Naomi Halachmi (iD) http://orcid.org/0000-0002-1409-3303
Ella Preger-Ben Noon (iD) http://orcid.org/0000-0001-5349-1112
Adi Salzberg (iD) http://orcid.org/0000-0002-0427-9809

## Decision letter and Author response

Decision letter https://doi.org/10.7554/eLife.70833.sa1
Author response https://doi.org/10.7554/eLife.70833.sa2

# Additional files

## Supplementary files

• Supplementary file 1. Results of the 1YH screen. This table summarizes the sequencing data of 146 positive clones identified in the 1YH screen.

• Supplementary file 2. Transgenic constructs used in this study. This table lists all the *dei*-related transgenes used in this study, including the vectors used and landing sites. The sequences of the wildtype and mutated dei$^{ChO-262}$ fragments are shown. The associated Figures are indicated.

• Supplementary file 3. Oligos and proteins used in the Electro Mobility Shift Assays. The Oligos and Protein Concentrations Used in the described Electro Mobility Shift Assays performed in this study. The mutated nucleotides in the competitor oligos are underlined.

• Transparent reporting form

## Data availability

All data generated or analysed during this study are included in the manuscript and supporting files. Source data files have been provided.

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
