## [Editor Report]

The study provides compelling evidence for a gene regulatory network involved in generating different sensory cell types from a common progenitor. The careful work shows how an enhancer can integrate the antagonistic relationship between two transcription factors for *Drosophila* sensory system development.

---

## [Decision Letter]

**Decision letter after peer review:**

Thank you for submitting your article "delilah, prospero and D-Pax2 constitute a gene regulatory network essential for the development of functional proprioceptors" for consideration by *eLife*. Your article has been reviewed by 3 peer reviewers, and the evaluation has been overseen by a Reviewing Editor and Utpal Banerjee as the Senior Editor. The following individuals involved in review of your submission have agreed to reveal their identity: Sonia Sen (Reviewer #2).

Essential revisions:

The three reviewers found the work to be of quality but they have identified a few serious issues on three points.

The first one, shared by reviewers #1 and #3 is the significance of the findings that would gain by being more general. Of course, we do not want you to start new experiments on other systems, but you should discuss in more depth how the GRN and the regulations can apply to other sensory organs or to other systems.

There is a point that needs to be addressed and was raised by reviewer #3 about the nature of the enhancer deletion, to make sure that this does not affect splicing or other structural features of the transcript.

The binding sites for Pros and Pax2 appear to mediate regulation. However, you should show that you can really separate the binding of these proteins to the enhancer and that mutations in one site do not affect the other.

There is a list of other significant issues raised by the reviewers and you should try your best to address these suggestions.

*Reviewer #1 (Recommendations for the authors):*

In general, the paper expands on previous work by this group demonstrating that Pros and Pax2 play opposite roles on the dei ChO enhancer in "cousin" cell types. While well-executed, the overall impact of the study is rather limited, expecially given that a similar relationship has been observed in other development contexts. Providing additional mechanistic insight into Pros-Pax2 antagonism and discussion of these factors in broader contexts besides *Drosophila* sensory systems may increase impact.

*Reviewer #3 (Recommendations for the authors):*

(1) The yeast 1-hybrid screen is not well described. It is unclear what the authors mean by "two proteins were identified to bind the bait with high confidence" and what defines the other 3 candidates as being "moderate confidence"? The methods only describe that the "Aureobasidin A selection system" was used – with no description of what define high/moderate/low confidence. For those not familiar with this system the authors should clearly define in either the Results section or Methods how genes were defined as high/moderate/low confidence, how the results were validated, and how many independent clones of each gene were isolated – especially of D-Pax2/sv.

(2) In Figure 1E-1H, it is unclear what transgenes were used to study these enhancers, what location they are inserted into, and what stain is shown in the Figure. The Figure legend simply states embryonic expression patterns driven by each enhancer and the methods imply that the placZattb was used with a reference to Table S4. But I could only find one supplemental Table (the yeast 1-hybrid results). The authors need to (A) clearly state what transgenes were made in this study versus others, (B) state the explicit attP2 landing location used to insert each transgene, (C) state if each transgene was always inserted into the same or different landing locations for the comparative analysis, (D) state if similar results obtained using two different landing sites, and (E) state if all transgenic flies were studied as homozygotes (i.e. two copies of the transgene) or as heterozygotes, etc.

(3) In Figure 2J and 2L, the authors state that ectopic Pax2 induces extra Dei and deicho-lacZ expression – but it is unclear in how many cells? Given that Dei is expressed in other cell types – it is hard to tell how consistently Pax2 induces extra Dei expression. It would also be informative to know what specific cell types Pax2 is sufficient to induce Dei. Is Dei only induced by ectopic Pax2 within other cells of the ch organ lineage or does it also induce Dei in cells within other sensory organ lineages or even outside of the PNS? Addressing such questions would help to determine the sufficiency of DPax2 to induce this enhancer element.

(4) In Figure 4 – the authors need to provide more information and more controls when using and interpreting the data of the fly line with the 307bp enhancer deletion generated by genome editing. Since this deletion is located within an intron of the gene – and very close to the intron/exon boundary between intron 1 and exon2 – the authors need to both clearly highlight (A) where the intron/exon boundaries are within dei; (B) the exact location of the 307bp deletion; and (C) the authors should experimentally demonstrate that the dei transcript in the enhancer mutant fly line is not disrupted in any way that impacts the making of the Dei protein. This data is essential, as the authors conclude that failure to express the Dei protein in the cap/attachment cells is what leads to defective larval movement. However, if the intronic deletion also alters mRNA splicing and results in an abnormal transcript followed by an abnormal protein – then interpreting the phenotypes generated by these mutants is difficult given that Dei is not only expressed in the cap/attachment cells but also within tendon cells that could also impact *Drosophila* movement.

(5) In the EMSA methods, the authors site a Table SX with all the protein concentrations used in the in vitro DNA binding studies – but I could not find any table with such information.

(6) In Figure 5 – The authors highlight that the Pros site 1 and dPax2 site 1 overlap each other and the authors subsequently test a mutation for Pros site 1 and Pax2 site 1 in the reporter assays. However, it is unclear which specific mutations were tested in the reporter assays and importantly IF the Pros site 1 mutation was specific to Pros and does not impact Pax2 binding and IF the Pax2 site 1 mutation was specific to Pax2 and does not impact Pros binding. The authors need to explicitly state if each in vivo mutation was designed to be specific to either Pax2 or Pros binding or if the mutations impact the binding of both factors. Thus, it would be very helpful, if the authors showed the mutation sequence tested for each site listed under the sequence confirmation in panel B and use a name for each mutation that matches the mutations tested in the Supplemental Figures.

(7) With the exception of transgenic reporter expression in Figures 5M and 5N and the larvae movement assays, there is very little quantification of phenotypes throughout the paper. In point #4 above, I already mentioned one such example where Pax2’s impact on Dei and Dei enhancer activity could be quantified. While it may not be possible to quantify every type of expression changes studied throughout the manuscript, the authors should consider if it is possible to quantify the impact of Pros on Dei expression/Dei enhancer activity in a similar manner.

(8) To better understand the variability in DNA binding sites for Pros and Pax2, it would be helpful to align each of the Pax2 and Pros sites identified within the dei enhancer with each other and under the published consensus PWM motifs for Pax2 and Pros respectively. Such information would be very helpful given that the authors stated the identified sites, especially for Pros do not match the consensus motifs well.

(9) In Figure 2, the authors use two different sv alleles but do not describe either one. Are these alleles protein null alleles (no Pax2 protein detected) or hypomorphic? Moreover, why show the impact on Dei using the sv6 allele in panels A/B, but then show the impact on aTub85E in sv7 alleles in C/D? Were similar results observed in both? Would be better to show the data for one allele in Figure 2 and then include a supplemental figure with the other and explicitly state whether each give identical or different severities of phenotypes – ideally with quantitation and statistical analysis.

(10) The presentation of the results, especially the first few sections, made it difficult to assess what data was based on previous papers and what data was new in this study. For example, the first paragraph of the results describes the mapping of the minimal core dei enhancer (shown in Figure 1D through 1H) and the second paragraph describes an RNAi screen identifying shaven (DPax2) as a key regulator of this enhancer (not shown). However, both findings were previously published by the Salzberg lab – and thus while I found Figure 1 to be helpful to understand the paper, the authors should consider presenting these findings in the Introduction or only briefly summarized at the beginning of the results in a single statement or two.

(11) It would be more convincing if the authors quantify the number of ectopic Dei positive cells within an EnGal4 stripe vs between the EnGal4 stripes in the control (UAS-GFP vs UAS-sv/UAS-GFP) animals – which would determine autonomous vs non-autonomous impacts on Dei expression. In addition, the authors could use additional marker genes of specific cell fates to determine if Pax2 induces Dei in only ch organ lineages, in other sensory cell lineages, and/or in cells outside the PNS.

(12) The authors should use RT-PCR followed by sequencing to make sure the transcripts of the dei mRNA are properly spliced and not disrupted in flies homozygous for the deletion.

---

## [Author Response]

Essential revisions:The three reviewers found the work to be of quality but they have identified a few serious issues on three points.The first one, shared by reviewers #1 and #3 is the significance of the findings that would gain by being more general. Of course, we do not want you to start new experiments on other systems, but you should discuss in more depth how the GRN and the regulations can apply to other sensory organs or to other systems.

The role of opposing Pros and Pax2 activities in dictating cell specific differentiation programs in sensory organs has been previously documented (as thoroughly discussed in pages 14-16). However, to the best of our knowledge, our work identifies the first enhancer which serves as a molecular platform for integrating these opposing signals. While the identified *dei^ChO-262^* enhancer is ChO-specific, it is reasonable to assume that other enhancers that regulate gene expression in other types of sensory organs encode coupled Pros and D-Pax2 binding sites. For example, the expression of *dei* itself in other (non-ChO) organs is regulated via different enhancers (as described in Nachman et al., 2015). Some of these enhancers are responsible for regulating *dei* expression in tissues where Pros and Pax2 play opposing roles, such as the eye and wing margin ES organs (the *dei^wing+eye^* enhancer; Nachman et al., 2015). It is beyond the scope of this work, but in the future it will be interesting to decipher whether these enhancers also integrate opposing effects of Pax2 and Pros. These points are now included in the Discussion. To address reviewer’s #3 comment regarding the generality of the *dei^ChO-262^* function in other ChOs, we tested the function of the wild type and mutated *dei^ChO-262^* enhancer in the adult leg ChOs and found that the regulatory interactions within this GRN are conserved in the adult ChO lineages. This new data is presented in Figure 5—figure supplement 5. In addition, we highlight better the similar trends evident in other type of larval ChOs (Lch1, VChA/B) as pointed in Figures 2, 3, 5 and Figure 3—figure supplement 1.

There is a point that needs to be addressed and was raised by reviewer #3 about the nature of the enhancer deletion, to make sure that this does not affect splicing or other structural features of the transcript.

This point has been addressed and is discussed in detail in our response to comment #4 of reviewer #3.

The binding sites for Pros and Pax2 appear to mediate regulation. However, you should show that you can really separate the binding of these proteins to the enhancer and that mutations in one site do not affect the other.

In the revised manuscript we elaborate on the issue of overlap between Pros and D-pax2 binding sites. The introduced changes are described in detail in our response to comment #6 of reviewer #3.

Reviewer #3 (Recommendations for the authors):1) The yeast 1-hybrid screen is not well described. It is unclear what the authors mean by "two proteins were identified to bind the bait with high confidence" and what defines the other 3 candidates as being "moderate confidence"? The methods only describe that the "Aureobasidin A selection system" was used – with no description of what define high/moderate/low confidence. For those not familiar with this system the authors should clearly define in either the Results section or Methods how genes were defined as high/moderate/low confidence, how the results were validated, and how many independent clones of each gene were isolated – especially of D-Pax2/sv.

Following the reviewer’s comment we have significantly expanded the description of the Y1H screen in the Results section and, mainly, in the Methods section. We now describe in greater details the constructs used, the use of Aureobasidin-A as a selection system, the number of clones screened, and number of clones sequenced (which is also shown in the Figure 1—figure supplement 1 and Supplementary File 1). We also describe what defines high/moderate/low confidence (see pages 6 and 21-22 and the legend to Figure 1—figure supplement 1). The relevant references were added to the manuscript.

2) In Figure 1E-1H, it is unclear what transgenes were used to study these enhancers, what location they are inserted into, and what stain is shown in the Figure. The Figure legend simply states embryonic expression patterns driven by each enhancer and the methods imply that the placZattb was used with a reference to Table S4. But I could only find one supplemental Table (the yeast 1-hybrid results). The authors need to A) clearly state what transgenes were made in this study versus others, B) state the explicit attP2 landing location used to insert each transgene, C) state if each transgene was always inserted into the same or different landing locations for the comparative analysis, D) state if similar results obtained using two different landing sites, and E) state if all transgenic flies were studied as homozygotes (i.e. two copies of the transgene) or as heterozygotes, etc.

The reporter constructs shown in Figure 1E-H were cloned into the pG-Pelican vector and transgenic strains were generated using *P*-element transformation. For each construct, independent transgenic strains with insertions on the X, 2^nd^ and 3^rd^ chromosomes were isolated and tested (described in Nachman et al., 2015). The *dei^ChO-262^* reporter construct was later generated again using the *pLacZ-attB* vector and injected into the *attP2* site (*attP2* is the landing site located at 68A4), into which all the mutated reporter constructs were inserted as well. All the transgenic reporters have been studied as homozygotes. Supplementary File 2 (and the Star methods table) summarizes all the transgenes, vectors used and landing sites when known. We apologize for the loss of this table from the previous version of the manuscript.

3) In Figure 2J and 2L, the authors state that ectopic Pax2 induces extra Dei and deicho-lacZ expression – but it is unclear in how many cells? Given that Dei is expressed in other cell types – it is hard to tell how consistently Pax2 induces extra Dei expression. It would also be informative to know what specific cell types Pax2 is sufficient to induce Dei. Is Dei only induced by ectopic Pax2 within other cells of the ch organ lineage or does it also induce Dei in cells within other sensory organ lineages or even outside of the PNS? Addressing such questions would help to determine the sufficiency of DPax2 to induce this enhancer element.

It is very difficult to determine unambiguously how many cells express *dei* or the *dei^cho^-lacZ* reporter ectopically upon Pax2 expression. The main reason is that ectopic expression of Pax2 has detrimental effects on the pattern of ChO migration and on other morphogenetic processes in the embryo. Thus, it is very difficult to identify cells based on their location and to some extent on the expression of other markers (for example, aTub85E is normally restricted to ChO cells, but upon ectopic Pax2 expression it is expressed in epidermal cells). However, clearly, cells outside the PNS express the *dei* gene and the *dei^cho^-lacZ* reporter upon Pax2 ectopic expression. As seen in Figure 2J and L, the expression of *dei^cho^-lacZ* is induced in the majority of epidermal cells within the *en* domain (labeled with GFP) that do not express it under normal conditions. The expression of Dei itself and the ChO marker aTub85E is also evident in many of these cells. In the revised manuscript we refer more clearly to this point (page <milestone-start /> <milestone-end /> 7 and the legend to Figure 2).

4) In Figure 4 – the authors need to provide more information and more controls when using and interpreting the data of the fly line with the 307bp enhancer deletion generated by genome editing. Since this deletion is located within an intron of the gene – and very close to the intron/exon boundary between intron 1 and exon2 – the authors need to both clearly highlight (A) where the intron/exon boundaries are within dei; (B) the exact location of the 307bp deletion; and (C) the authors should experimentally demonstrate that the dei transcript in the enhancer mutant fly line is not disrupted in any way that impacts the making of the Dei protein. This data is essential, as the authors conclude that failure to express the Dei protein in the cap/attachment cells is what leads to defective larval movement. However, if the intronic deletion also alters mRNA splicing and results in an abnormal transcript followed by an abnormal protein – then interpreting the phenotypes generated by these mutants is difficult given that Dei is not only expressed in the cap/attachment cells but also within tendon cells that could also impact *Drosophila* movement.

We fully agree with this comment and followed the reviewer’s suggestion. To verify that deleting the intronic enhancer does not affect splicing or other structural features of the transcript, cDNA was synthesized from homozygous *dei^DChO^* and control flies. A 416 bp fragment was amplified by PCR from the cDNA samples using primers located on both sides of the intron (in the 1^st^ and 2^nd^ exons). Sequencing of the PCR products verified the presence of normally structured *dei* transcript in the *dei^DChO^* mutant. We have now added this new experimental data to the manuscript together with more detailed presentation of the gene structure (result section – page 9; Materials and methods, and Figure 3—figure supplement 2).

5) In the EMSA methods, the authors site a Table SX with all the protein concentrations used in the in vitro DNA binding studies – but I could not find any table with such information.

We thank the reviewer for drawing our attention to this missing information. We have now added the missing table (Supplementary File 3). The table includes the sequence of the oligos used in the different EMSAs along with the concentrations of the proteins and the associated Figures. We also included a table that summarizes all the transgenic lines used in this study (Supplementary File 2).

6) In Figure 5 – The authors highlight that the Pros site 1 and dPax2 site 1 overlap each other and the authors subsequently test a mutation for Pros site 1 and Pax2 site 1 in the reporter assays. However, it is unclear which specific mutations were tested in the reporter assays and importantly IF the Pros site 1 mutation was specific to Pros and does not impact Pax2 binding and IF the Pax2 site 1 mutation was specific to Pax2 and does not impact Pros binding. The authors need to explicitly state if each in vivo mutation was designed to be specific to either Pax2 or Pros binding or if the mutations impact the binding of both factors. Thus, it would be very helpful, if the authors showed the mutation sequence tested for each site listed under the sequence confirmation in panel B and use a name for each mutation that matches the mutations tested in the Supplemental Figures.

We thank the reviewer for drawing our attention to this missing information. We have now revised Figure 5 to include the mutated nucleotides that were tested in transgenic flies. In addition, we added a Supplementary Figure (Figure 5—figure supplement 3) and Table (Supplementary File 2) that presents the full sequence of all the mutants used for the in vivo assays.

Regarding the overlap between Pros site1 and D-Pax2 site 1: The mutation in Pax2 site 1 was specific to the Pax2 site and did not affect Pros binding to the Pros site 1 in EMSA. As shown in Figure 5—figure supplement 1D, mutH in probe 3b, which is equivalent to the in vivo mutation in Pax2 site 1, was able to compete the WT probe for the binding to Pros. The mutation in Pros site 1, on the other hand, did affect the binding of Pax2 to Pax2 site 1 in vitro*.* As shown in Figure 5—figure supplement 1D, mutE in probe 3b, which is equivalent to the in vivo mutation in Pros site 1, failed to compete the WT probe for the binding Pax2. As mutations in Pax2 site 1 did not affect the expression of the reporter gene in cap cells on its own (Figure 5M), we do not anticipate that the mutation in Pros site 1 will have a major effect on the function of *dei^ChO-262^*. Saying that, it is possible that if Pax2 binding was not affected by the mutation in Pros site 1, we would observe even higher elevation in the expression levels of the reporter in scolopale cells of this line (Figure 5N). We have revised the manuscript to address this concern (page 12).

7) With the exception of transgenic reporter expression in Figures 5M and 5N and the larvae movement assays, there is very little quantification of phenotypes throughout the paper. In point #4 above, I already mentioned one such example where Pax2’s impact on Dei and Dei enhancer activity could be quantified. While it may not be possible to quantify every type of expression changes studied throughout the manuscript, the authors should consider if it is possible to quantify the impact of Pros on Dei expression/Dei enhancer activity in a similar manner.

While such quantifications could be desirable, we feel that they are not necessary because the effects of Pros and Pax2 on *dei* expression shown in Figures 2-4 are discussed in terms of *spatial* distribution rather than *level* of expression. At this point, quantifying the level of expression in the different genotypes is impossible since only in the experiments shown in Figure 5 all the samples were imaged using the exact same imaging parameters thus allowing for quantitative comparisons. It is important to note that the representative pictures showed in the manuscript represent a large number of examined segments and that the observed phenotypes were fully penetrant. For example, we have examined 116 segments of *pros^17^* mutant embryos, all of them showed the described ectopic expression of *dei*.

8) To better understand the variability in DNA binding sites for Pros and Pax2, it would be helpful to align each of the Pax2 and Pros sites identified within the dei enhancer with each other and under the published consensus PWM motifs for Pax2 and Pros respectively. Such information would be very helpful given that the authors stated the identified sites, especially for Pros do not match the consensus motifs well.

We thank the reviewer for this good advice. We have added Figure 5—figure supplement 4 that shows the suggested alignments, which are also discussed in the text (in the results -page 11 and the Discussion – page 16).

9) In Figure 2, the authors use two different sv alleles but do not describe either one. Are these alleles protein null alleles (no Pax2 protein detected) or hypomorphic? Moreover, why show the impact on Dei using the sv6 allele in panels A/B, but then show the impact on aTub85E in sv7 alleles in C/D? Were similar results observed in both? Would be better to show the data for one allele in Figure 2 and then include a supplemental figure with the other and explicitly state whether each give identical or different severities of phenotypes – ideally with quantitation and statistical analysis.

In the revised version of the manuscript, we present in Figure 2 a single *sv* allele (*sv^6^*) which is a protein null allele.

10) The presentation of the results, especially the first few sections, made it difficult to assess what data was based on previous papers and what data was new in this study. For example, the first paragraph of the results describes the mapping of the minimal core dei enhancer (shown in Figure 1D through 1H) and the second paragraph describes an RNAi screen identifying shaven (DPax2) as a key regulator of this enhancer (not shown). However, both findings were previously published by the Salzberg lab – and thus while I found Figure 1 to be helpful to understand the paper, the authors should consider presenting these findings in the Introduction or only briefly summarized at the beginning of the results in a single statement or two.

We fully agree with this comment and revised the first paragraph of the Results section (pages 5-6) accordingly (without moving Figure 1). We hope that the distinction between old and new data is much clearer now.

11) It would be more convincing if the authors quantify the number of ectopic Dei positive cells within an EnGal4 stripe vs between the EnGal4 stripes in the control (UAS-GFP vs UAS-sv/UAS-GFP) animals – which would determine autonomous vs non-autonomous impacts on Dei expression. In addition, the authors could use additional marker genes of specific cell fates to determine if Pax2 induces Dei in only ch organ lineages, in other sensory cell lineages, and/or in cells outside the PNS.

It is not clear to us why the question of autonomous versus non-autonomous effect of Pax2 on Dei expression is raised as none of the experimental results support a non-autonomous effect. Specifically for the induction of Pax2 expression under the regulation of *en-gal4*, such analysis is tricky because the ChO cells are born within the *en* domain and then migrate out to a more anterior compartment. The pattern of cell migration is disrupted when Pax2 is expressed, making the analysis of autonomous vs non-autonomous effects quite impossible. The data in Figure 2J, L demonstrates unambiguously that Pax2 can induce the ectopic expression of *dei* outside the PNS.

12) The authors should use RT-PCR followed by sequencing to make sure the transcripts of the dei mRNA are properly spliced and not disrupted in flies homozygous for the deletion.

We have followed the reviewer’s recommendation and used RT-PCR followed by sequencing to verify the correct structure of the *dei* transcript in the enhancer deletion mutant. The results are described in the text and in Figure 4—figure supplement 1.